# Tyrosyl-tRNA synthetase has a noncanonical function in actin bundling

Biljana Ermanoska [1,2,3], Bob Asselbergh [4,5], Laura Morant [1,2], Maria-Luise Petrovic-Erfurth[1,2], Seyyedmohsen Hosseinibarkooie [6,15], Ricardo Leitão-Gonçalves [1,2,16], Leonardo Almeida-Souza[1,2,17], Sven Bervoets [1,2,18], Litao Sun [7,19], LaTasha Lee[8,20], Derek Atkinson [1,2,21], Akram Khanghahi[1,2], Ivaylo Tournev[9,10], Patrick Callaerts[11], Patrik Verstreken[12,13], Xiang-Lei Yang [7], Brunhilde Wirth [6], Avital A. Rodal [3], Vincent Timmerman [2], Bruce L. Goode[3], Tanja A. Godenschwege [8] & Albena Jordanova [1,2,14] ✉

Dominant mutations in tyrosyl-tRNA synthetase (YARS1) and six other tRNA ligases cause Charcot-Marie-Tooth peripheral neuropathy (CMT). Loss of aminoacylation is not required for their pathogenicity, suggesting a gain-of-function disease mechanism. By an unbiased genetic screen in *Drosophila*, we link YARS1 dysfunction to actin cytoskeleton organization. Biochemical studies uncover yet unknown actin-bundling property of YARS1 to be enhanced by a CMT mutation, leading to actin disorganization in the *Drosophila* nervous system, human SH-SY5Y neuroblastoma cells, and patient-derived fibroblasts. Genetic modulation of F-actin organization improves hallmark electrophysiological and morphological features in neurons of flies expressing CMT-causing YARS1 mutations. Similar beneficial effects are observed in flies expressing a neuropathy-causing glycyl-tRNA synthetase. Hence, in this work, we show that YARS1 is an evolutionary-conserved F-actin organizer which links the actin cytoskeleton to tRNA-synthetase-induced neurodegeneration.

Protein translation is a fundamental cellular process in which aminoacyl-tRNA synthetases play a major role. These ubiquitous enzymes are responsible for the continuous and correct charging of tRNAs with their cognate amino acids during protein biosynthesis (canonical function)[1]. Dominant mutations in tyrosyl-tRNA synthetase (YARS1) and six other synthetases cause different subtypes of Charcot-Marie-Tooth disease (CMT)[2–4], the most common and currently incurable inherited peripheral neuropathy[5]. The pathology is mostly restricted to the axons of the peripheral nerves that degenerate in a length-dependent, dying-back fashion upon disease progression. How monoallelic genetic defects in such ubiquitous enzymes can cause selective damage of the peripheral nerves is not fully understood. Extensive evidence from us and others demonstrates that, at least for YARS1, glycyl-tRNA synthetase (GARS1) and alanyl-tRNA synthetase (AARS1), the CMT phenotype cannot be ascribed to a simple loss of

aminoacylation (reviewed in refs. [2,3]). Rather, a growing list of diverse noncanonical functions is being described for these enzymes[2,6,7]. CMT mutations could disrupt those non-aminoacylation activities or generate completely novel neurotoxic properties of the affected synthetases.

To search for commonalities in the mode of action of the mutant synthetases, systematic structural studies have been performed. Notably, the CMT mutations in GARS1[8], YARS1[9], HARS1[10], and AARS1[11], cause 3D-conformational opening that exposes sequences buried within the wild-type enzymes, thereby impacting their interaction properties. The structural relaxation of mutant GARS1 and AARS1 leads to a neomorphic interaction with the transmembrane receptor Neuropilin 1[8,11], while the conformational opening caused by three different CMT mutations in YARS1 increases the binding affinity to TRIM28[12], compared with the wild type proteins. An overarching and potentially

common mechanistic hypothesis explaining the neuropathology shared by these enzymes is currently lacking and this is hampering the development of effective therapeutics.

In this study, we focused on YARS1, a cytoplasmic tRNA ligase with additional nuclear roles in transcriptional regulation and response to oxidative stress[13], as well as extracellular signaling[14]. All five known CMT-causing mutations are in the catalytic domain of the protein[2,3], and yet YARS1[E196K] does not impair the enzyme kinetics[12]. Notably, when expressed in *Drosophila*, YARS1[E196K]— similarly to the aminoacylation-compromised YARS1[G41R] and YARS1[153-156delVKQV]— induces phenotypes recapitulating hallmark features of the human pathology, including motor deficits, electrophysiological neuronal dysfunction and axon terminal degeneration[15]. In the present study, we performed an unbiased genetic screen in the *Drosophila* model and together with in cellulo interactomics linked YARS1 dysfunction to the actin cytoskeleton organization. Further, we observed that YARS1 directly binds and bundles F-actin filaments in vitro, and induces cytoskeletal rearrangements in CMT-relevant paradigms.

The actin cytoskeleton is a major part of the cellular backbone, defining cell shape, dynamics, and internal organization, and is composed of actin filaments (F-actin) resulting from ATP-dependent reversible polymerization of globular actin (G-actin)[16]. Individual F-actin filaments are crosslinked into higher-order structures, such as bundles and branched networks, which drive a wide range of cellular processes, including cell motility, cell adhesion, intracellular transport, endocytosis, and cytokinesis. These processes require the precisely choreographed assembly, rearrangement, and turnover of actin networks, which is achieved by the concerted actions of a multitude of actin-binding proteins (ABPs). In neurons, actin rearrangements drive many processes, including neurite outgrowth and sprouting, vesicle dynamics, as well as stability or instability of the synapse[17]. Emerging evidence implicates regulation of the actin cytoskeleton as a causative and dysregulated process in neuronopathies. Altered actin dynamics is observed in experimental models of amyotrophic lateral sclerosis (ALS) and spinal muscular atrophy (SMA)[17], and increased profilin-2 levels are detected in iPSC-motoneuron models of different CMT subtypes[18]. Moreover, human genetic studies show that mutations in actin-regulatory proteins (profilin-1, alsin) cause subtypes of ALS, while actin-binding and/or bundling proteins (plastin-3, coronin) are high-risk factors or suppressors of ALS and SMA, respectively[18,19].

In this work, we report actin binding and bundling functions for YARS1, and demonstrate that they are relevant for CMT pathogenesis, as genetic manipulations of the actin cytoskeleton rescue cellular hallmarks of the neuropathy in *Drosophila*. Furthermore, we demonstrate that F-actin binding is a common feature for other disease-implicated synthetases and its manipulation can be beneficial for GARS1 neuropathy. Our findings support a common molecular mechanism of neurodegeneration involving the actin cytoskeleton.

## Results

### Genetic screen identifies actin-related YARS1[CMT] interactors

To identify the mechanistic basis of YARS1[CMT] toxicity, we performed an unbiased genetic modifier screen in *Drosophila*, as retinal (*GMR-Gal4*) expression of the enzymatically active *YARS1[E196K]* mutant makes fly eye ommatidia sensitive to disorganization. This sensitivity offers a robust, simple, and high-throughput test, which has been widely used for discovery of genetic modifiers of neurodegenerative diseases in *Drosophila*[20]. Notably, there is a major overlap of neurodegenerative pathways eliciting toxicity in the fly's developing eye and postmitotic neurons[21] and we have previously demonstrated that this read-out enables the identification of CMT-relevant phenotypic enhancers[9]. Using the adult fly eye as a screening platform, we systematically examined 557 *Drosophila* lines containing enhancer-promoter *P*-element insertions (*EP*s)[22–25] on the X-chromosome (Fig. 1a). The combination of *EP*s, carrying Gal4-binding sites and a Gal4-source provides a

modular system that will determine the expression of any gene downstream of the EP insertion site[22]. Depending on the genomic insertion, an *EP* could overexpress the downstream gene in the presence of a Gal4-source, misexpress partial products of it, or induce a hypomorph if the insertion disrupts the gene coding sequence. Many of the *EP*s (72.3%) we screened were inserted in proximity of genes encoding proteins belonging to different functional classes (Supplementary Fig. 1b, c, Supplementary Data 1). 96% of the screened *EP*s did not affect ommatidial organization (Supplementary Fig. 1a). We excluded from further analysis *EP*s that gave retinal phenotypes on their own (2.15%), induced lethality (0.36%) or had nonspecific, comparable effects when co-expressed with either *YARS1[E196K]* or *YARS1[WT]* (1.26%) (Supplementary Fig. 1a). Ultimately, we detected a strong rough eye phenotype upon *GMR-Gal4*-driven co-expression of *YARS1[E196K]* and the *BDSC_14274* (*Fim[EP]*) line, unlike the minor ommatidial disorganization when *Fim[EP]* was expressed alone or together with *YARS1[WT]* (Fig. 1c). *Fim[EP]* maps upstream of the gene Fimbrin (CG8649/*Fim*), in an orientation that will promote the expression of *Fim* and not the neighboring CG5445, as evidenced by the up-regulation of its three splice-isoforms in the presence of a Gal4 source (Supplementary Fig. 2a, b). Next, we tested all publicly available *EP*s inserted in the direction of *Fim* expression at different locations of the gene (see Methods) and found an additional *EP* inserted in *Fim* that induced mild ommatidial disorganization when co-expressed specifically with *YARS1[E196K]* – *BDSC_19171* (*Fim[XP]*) (Supplementary Figs. 1d, 2a). Thus, we identified two independent *EP*s inserted in the *Fim* gene that are mutant-specific *YARS1* genetic enhancers.

Fim is the sole, highly conserved *Drosophila* orthologue of a family of actin binding and -bundling proteins, which in humans are encoded by three Plastin (PLS) genes with distinct expression patterns (Supplementary Fig. 3). To confirm the interaction with *Fim/PLS*, we generated UAS-transgenes of full-length *Drosophila Fim* (upregulating all *Fim* isoforms), as well as of human *PLS2* and *PLS3* (Supplementary Fig. 2c). Retinal co-expression of *Fim* (Fig. 1d), *PLS2* (Fig. 1e) or *PLS3* (Fig. 1f) with *YARS1[E196K]* induced a rough eye phenotype, while all were neutral in the *YARS1[WT]* background. Thus, overexpression of either *Drosophila* or human versions of the actin bundler *Fim/PLS* enhanced the *YARS1[E196K]*-associated eye phenotypes in *Drosophila*.

To further explore the link between YARS1 and the F-actin-binding genetic interactor, we tested the modification effect of 23 candidates from a network of physically and genetically interacting partners of Fim and YARS1 (DroID database[26]), belonging to various functional classes (Supplementary Fig. 1e). We found that up-regulation of two additional actin-binding proteins (Coro and its close homolog dPod1*)* or loss of a regulator of F-actin bundling in flies (the IKKε kinase)[27] caused rough eye phenotypes when co-expressed with *YARS1[E196K]* (Supplementary Fig. 1f).

Independently, PLS3 and other actin binding proteins, including other F-actin bundlers, were captured in immunoprecipitation experiments in human (HEK293) cells expressing FLAG-YARS1[WT], but not the FLAG-tag alone (Supplementary Fig. 4a, b, Supplementary Data 2). We tested the interaction with PLS3 and α-actinin in vitro using recombinant proteins and found no direct binding with YARS1 (Supplementary Fig. 4c, d), suggesting that YARS1 and these ABPs might be constituents of a complex with another common interacting partner.

### YARS1 binds F-actin and organizes actin filaments in vitro

The recurrent association between YARS1 and proteins with actin-binding properties prompted us to explore whether the synthetase might directly bind actin. Initially, we tested the capacity of purified recombinant YARS1 to bind actin filaments in well-established co-sedimentation in vitro assays. Variable concentrations of pre-assembled F-actin were incubated with 0.5 μM recombinant YARS1[WT] or YARS1[E196K], and actin filaments were subsequently pelleted by high-speed centrifugation. In the absence of F-actin, YARS1 fractionated

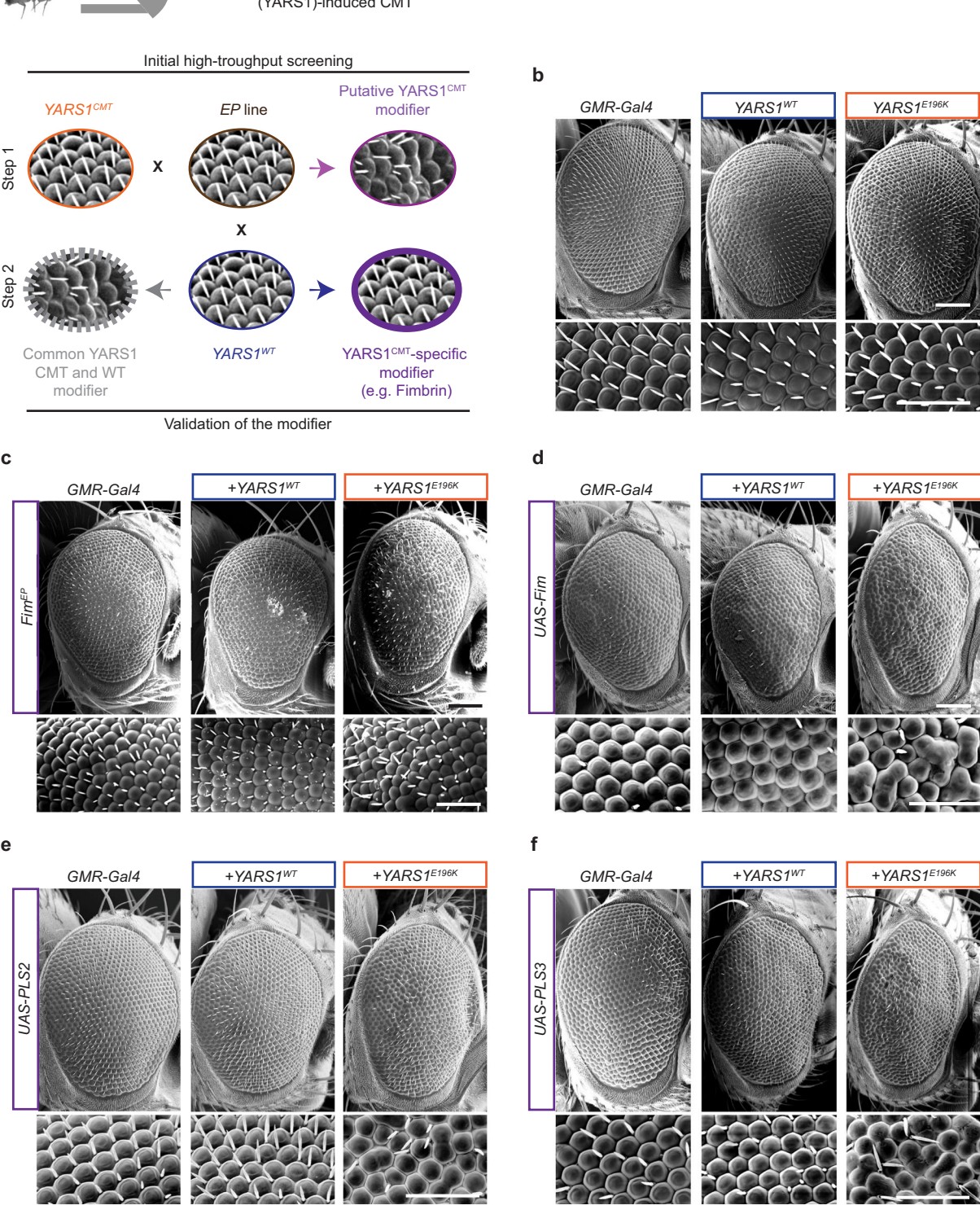

**Fig. 1 | A genetic screen in *Drosophila* identified actin cytoskeleton-related YARS1[CMT] interactors. a** A schematic representation of the principles of a genetic screen in which ommatidial disorganization of eyes in adult fruit flies was used as readout. *GMR-Gal4 > YARS1[CMT]* expressing flies (orange) were crossed to individual *EP* lines (brown). Genuine YARS1[CMT] modifier (purple) would induce disorganized ommatidia only in the mutant background and will not have an effect when crossed to YARS1[WT]-expressing flies in a validation step (blue). See also a detailed description in the Methods. Scanning electron micrographs of adult fly eyes in **b** controls, and **c** upon co-expression of *Fim[EP]*. Assessment of interaction with transgenic *UAS-Fim* (**d**), and the human orthologs *UAS-PLS2* (**e**) and *UAS-PLS3* (**f**) with *YARS1[WT]* or *YARS1[E196K]*. The experiments were repeated at least three independent times with similar outcomes. Scale bars – 50 μm.

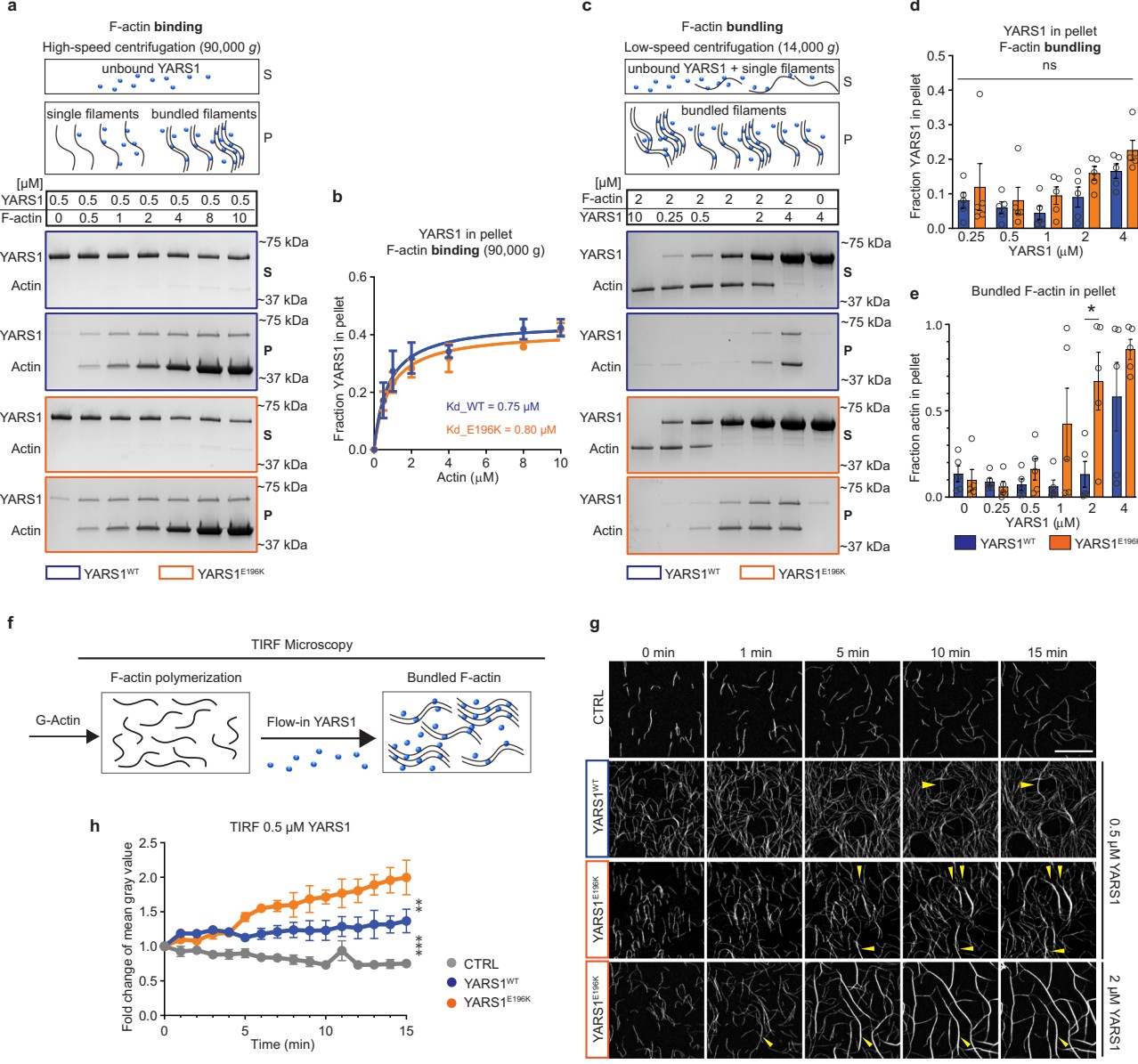

**Fig. 2 | YARS1 binds to F-actin and organizes actin filaments in vitro.**
**a** Coomassie-stained gels of supernatant (S) and pellet (P) fractions from high-speed pelleting (see schematic) of 0.5 μM YARS1[WT] or YARS1[E196K] upon increasing F-actin concentrations. **b** Graphical representation of the YARS1 fraction in the pellet obtained in three independent experiments, fitted into Michaelis-Menten nonlinear curve. **c** F-actin bundling capacity of YARS1 proteins tested upon low-speed pelleting (see schematic) of increasing concentrations of YARS1 and pre-assembled 2 μM F-actin. Graphs in **d** and **e** show the fraction of bundled F-actin and YARS1 in the pellets from five independent experiments; error bars in **b**, **d**, and **e**−SEM; *$p = 0.0187$, two-sided unpaired $t$ test. Supernatant and pellet samples derive from the same experiment and gels were processed in parallel. **f** A schematic

representation of the TIRF experiment. **g** Representative time-lapse images from a TIRF microscopy movie of 10% OG-actin labeled filaments alone (CTRL) or in the presence of YARS1 demonstrate stronger bundling capacity of YARS1[E196K] at 0.5 μM (areas of bundle formation depicted with yellow arrowhead) and pronounced cable formation at 2 μM. **h** Graph representing the fold change of mean gray value of OG-actin in control and upon flow-in of 0.5 μM YARS1, measured in areas of cable formation. $n = 3$ represents randomly selected, non-overlapping fields of view, imaged within one TIRF experiment per condition (CTRL, YARS1[WT], and YARS1[E196K]). Presented data derive from one of two independently performed experiments. Error bars−SD; **$p = 0.0011$, ***$p = 0.0003$, two-sided unpaired t-test. Source data are provided as a Source data file.

primarily in the supernatant. However, YARS1 shifted to the pellet fraction with increasing concentrations of F-actin (Fig. 2a), similar to an established actin-binding protein (Supplementary Fig. 5a), indicating that YARS1 binds directly to actin filaments. Based on the dissociation constants ($K_D$s) obtained from this experiment (Fig. 2b), we concluded that both YARS1[WT] and YARS1[E196K] have similar binding affinities for preassembled F-actin.

Next, we assessed whether the binding of YARS1 to actin has any functional impact on F-actin assembly. We performed in vitro F-actin polymerization in the presence of recombinant YARS1 followed by

differential centrifugation. High-speed centrifugation pulls down all F-actin filaments regardless of whether they are single filaments or organized into higher-order structures, while low-speed sedimentation pellets only larger, higher-order F-actin assemblies, including bundles. In our case, high-speed pelleting of the reaction mixture demonstrated that YARS1 did not change the steady-state ratio of G-actin (supernatant) vs. F-actin (pellet) (Supplementary Fig. 5b, c), suggesting it does not alter the critical concentration for actin assembly. Then, we investigated potential effects of YARS1 on higher-order F-actin organization by performing low-speed sedimentation

assays. Upon incubation of increasing concentrations of YARS1 proteins with 2 μM preassembled F-actin, both YARS1[WT] and YARS1[E196K] showed bundling capacity, demonstrated by the progressive shift of F-actin from the supernatant (single filaments) to the pellet fraction (bundled filaments) (Fig. 2c–e). Surprisingly, YARS1[E196K] induced F-actin bundling already at 1 μM, and progressively shifted F-actin to the low-speed pellet at higher concentrations, saturating its effects at 4 μM YARS1[E196K]. YARS1[WT] initiated a shift of F-actin to the pellet (i.e., bundled filaments) only at 2 μM. At 4 μM concentration both proteins comparably bundled most of the F-actin.

To further demonstrate the bundling activity of YARS1 in real time we used total internal reflection microscopy (TIRF) assays. Non-tethered Oregon Green (OG)-labeled actin filaments were grown in TIRF chambers, where their organization in higher order structures was imaged after a flow in of F-buffer (control) or YARS1 (see scheme in Fig. 2f). F-actin bundle formation was detected in this assay using 0.5 μM YARS1, and a stronger effect was observed for the same concentration of the mutant protein (Fig. 2f, g; Supplementary Movie 1). At a higher concentration (2 μM), both YARS1[WT] and YARS1[E196K] induced bundling, with most of the initial individual filaments merging into thicker F-actin cables (Fig. 2g; Supplementary Fig. 5f, g; Supplementary Movie 1). In control reactions (buffer alone), we observed no significant bundling of filaments in the 20 min observation window. Altogether, we demonstrate that YARS1 binds to F-actin and organizes actin filaments into tight long bundles. The concentration (and the associated $K_D$s) at which both wild type and mutant YARS1 bind to F-actin is in the range of known established bundlers[28,29]. This previously unknown property of YARS1 is stronger in the tested CMT-associated mutant.

To evaluate if actin-bundling affects the canonical tRNA ligase activity, we measured the aminoacylation activity of YARS1 in the presence of F-actin. No changes in the tRNA-charging activity of YARS1[WT] or YARS1[E196K] were detected in vitro (Supplementary Fig. 5h). These data suggest that at least at the concentrations tested, F-actin binding or bundling do not significantly interfere with the canonical role of YARS1.

Finally, since the actin-binding and bundling properties of YARS1 resemble the function of its genetic (but not physical) interactor PLS3, we asked whether they will influence each other's binding to actin. Indeed, the individual bundlers can be coming on and off filament sides while the group of molecules collectively maintains a stable bundle[30]. We preincubated 2 μM F-actin with 1 μM PLS3 at three different YARS1 concentrations, followed by high-speed co-sedimentation (Supplementary Fig. 5d, e). At the concentrations tested, we observed mild decrease of PLS3 in the filamentous actin-rich pellet, indicating that YARS1 and PLS3 might bind to F-actin in a competitive manner.

## YARS1[CMT] disrupts actin cytoskeleton organization in cellulo

To complement our functional genomics and in vitro biochemical findings, we turned to mammalian cell models. We first explored the association of YARS1 with the actin cytoskeleton by co-transfecting HeLa cells with YARS1-eGFP and the F-actin marker Lifeact-mCherry. We found YARS1 to be enriched at different F-actin assemblies, such as parts of the cell cortex at the cell periphery (Supplementary Fig. 6). Live cell imaging revealed YARS1 presence at highly dynamic protrusions at cell borders, resembling F-actin-rich structures (lamellipodia and associated membrane ruffles)[31] (Supplementary Fig. 6b, Supplementary Movie 2). Based on the presence of YARS1 at these dynamic actin-dependent protrusions, we next asked if YARS1[CMT] will modify their features. To this end, we examined fibroblasts from a control individual and a YARS1[E196K]-patient, expressing mutant YARS1 at endogenous levels. We plated the cells on micropatterns which provide the advantage of restricting the cell to adopt predetermined shape, controlling their substrate adhering compartments and the formation of

characteristic actin networks[32]. Control fibroblasts plated on Y-shaped micropatterns adopted a triangular shape and featured elongated F-actin structures (stress fibers representing unbranched, contractile F-actin) aligned with the triangular perimeter of the cells, as well as a branched actin meshwork at the cell apices[32] (Fig. 3a–d). In YARS1[E196K] fibroblasts, irregularities in the F-actin pattern were observed, ranging from absent or shorter stress fibers to aberrant apical and central F-actin accumulations. In both control and patient fibroblasts, immunolabeled endogenous YARS1 was distributed throughout the cytoplasm, along the stress fibers and at the apices enriched in F-actin accumulations, especially in the CMT cells (Fig. 3e). Thus, CMT patient-derived fibroblasts exhibit defects in actin cytoskeleton organization.

Next, we studied human SH-SY5Y neuroblastoma cells stably expressing YARS1[WT] or CMT-causing mutants (YARS1[E196K] and YARS1[G41R]) (Supplementary Fig. 7a–c). We assessed global migration properties of non-differentiated wild type and mutant cells, in response to a mechanical scratch in confluent cultures by following the scratch closure within 72 h (Fig. 4a). In line with potential actin cytoskeleton defects, we found significantly slower cell migration in the two CMT-mutant cell lines compared to YARS1[WT] in the first 24 h (Fig. 4b). To demonstrate that the impaired global migration is a result of discrete cellular defects in the YARS1[CMT] cell lines, we performed faster phase-contrast microscopy at higher magnification and examined the formation and dynamics of individual lamellipodia (Fig. 4c, d). Cells expressing YARS1[CMT] displayed shorter protrusions that were less persistent and featured increased protrusion rate compared to the control cells (Fig. 4e). Next, we examined for functional defects in differentiated SH-SY5Y neuron-like cells that feature primary neurites as well as secondary branches at day 8 of differentiation (Fig. 4f). Interestingly, we found that YARS1[CMT] cells develop significantly shorter primary neurites and have a smaller fraction of secondary branches compared to controls (Fig. 4g). Combined, these results show that in SH-SY5Y-derived neuron-like cells the YARS1[CMT] mutations impair the formation of protrusions known to heavily relay on functional actin cytoskeleton, and subsequent neurite outgrowth and branching.

## Fim modifies neuronal phenotypes in the fly YARS1[CMT] model

To explore putative actin rearrangements and associated neuronal phenotypes in vivo, we examined the larval neuromuscular junction (NMJ) of our YARS1[CMT] Drosophila model. NMJs are model synapses considered to be a "site of lesion" in the dying-back axonopathies, including CMT[33]. We previously showed that YARS1[CMT] induces an NMJ's undergrowth, characterized by decreased total NMJ length and number of presynaptic specializations (i.e., boutons)[9]. To look at neuronal actin cytoskeleton, we expressed the F-actin marker UAS-Lifeact (Lifeact)[34] pan-neuronally and observed F-actin to be enriched at the periphery of individual presynaptic specializations in control larvae. Either YARS1[WT] or YARS1[E196K] expression induced a distinct, more dispersed Lifeact signal (Fig. 5a, Supplementary Fig. 8a), comparable to overexpression of the actin bundler PLS3 (Supplementary Fig. 8b, c). Similarly, co-expression of GFP-tagged Actin (Act5C::GFP) with YARS1 and PLS3 demonstrated reduction of the number of presynaptic assemblies in boutons (Supplementary Fig. 8d, e). Thus, independent use of actin and an actin marker demonstrated changes in the distribution and organization of presynaptic actin cytoskeleton upon YARS1[CMT] expression aligned with the effect of the actin bundler PLS3.

At those NMJs, transgenic YARS1 was present in the boutons, as well as in the axonal compartment (Supplementary Fig. 9a, b), similarly to endogenous Fim (Supplementary Fig. 9e, f). Furthermore, immunohistochemical analysis of the sub-bouton organization identified YARS1 in proximity to synaptic vesicles (SVs), as evidenced by co-localization with the marker proteins Synaptotagmin 1 (Syt::eGFP) (Supplementary Fig. 9c) and Synapsin (Supplementary Fig. 11a).

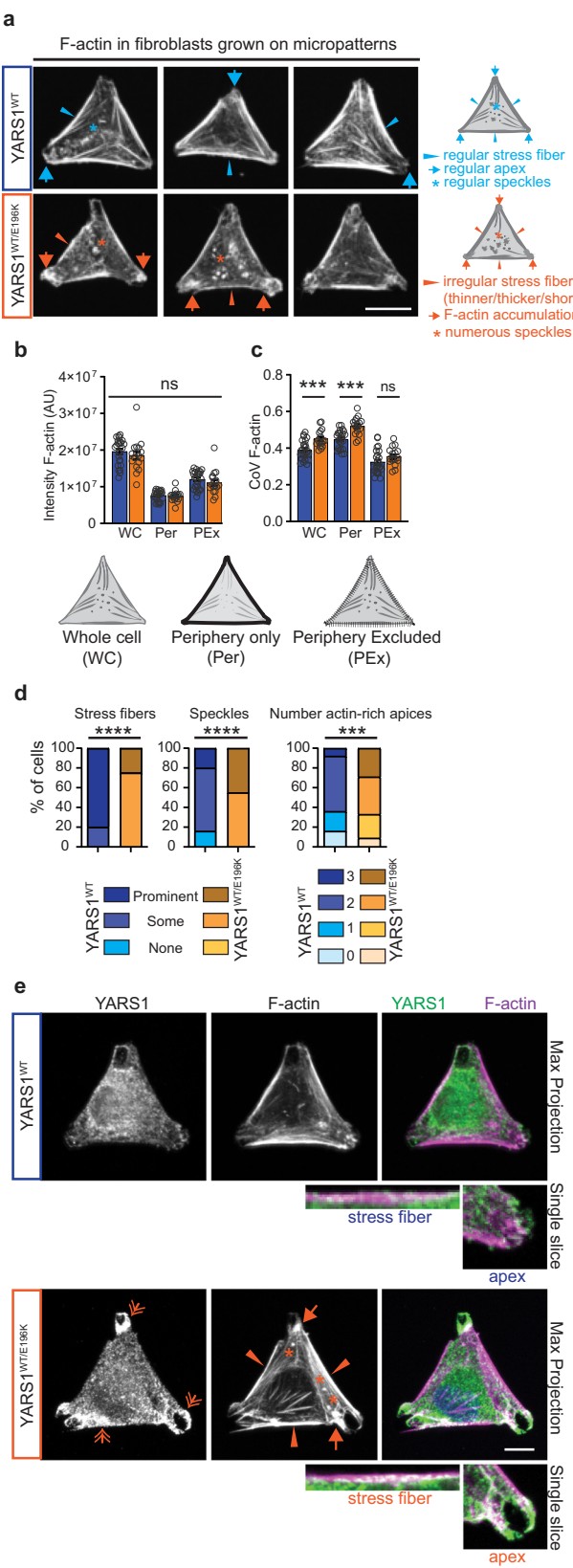

**Fig. 3 | Actin cytoskeleton organization is perturbed in YARS1$^{CMT}$ fibroblasts.**
**a** Fibroblasts from a control individual (*YARS1$^{WT/WT}$*) and a CMT patient
(*YARS1$^{WT/EI96K}$*), plated on fibronectin-coated Y-shaped micropatterns, adopt a tri-
angular shape. F-actin, stained with fluorescently labeled phalloidin, is organized
into linear stress fibers at the periphery (blue arrowheads) or crossing the center of
cells, and branched structures at the cell apices (blue arrows). Dysregulated stress
fibers (orange arrowheads), F-actin accumulations at the cell apices (orange
arrows), and speckled accumulations (orange asterisk) are observed in CMT
fibroblasts. A schematic of a representative cell with a description of the F-actin
structures is shown on the right side of the micrographs. **b** Quantification of the
fluorescence intensity, and **c** coefficient of variation (CoV) of the F-actin signal in
the whole triangle-shaped cell, and in cellular regions limited to the stress fiber-
occupied periphery or excluding it. **d** Frequency of actin-related phenotypes in
control and CMT fibroblasts. In **b**, **c**, and **d**, $n = 25$ (Control) and $n = 17$ (YARS1$^{E196K}$)
cells from one experiment out of two independently performed experiments.
**e** Maximum intensity projections of confocal images of immunolabeled YARS1
(green) and phalloidin-labeled F-actin (magenta) from control and mutant fibro-
blasts. Single slice insets of a stress fiber and an apex in the studied fibroblasts. The
experiments were repeated two times with similar outcomes. Scale bars – 20 μm.
Data in **b** and **c** are presented as mean values ± SEM and analyzed by two-sided
unpaired *t* test with ns – non-significant $p = 0.3035$ in **b**, $p = 0.1080$ in **c**;
***$p = 0.0006$. After a Chi-square test in **d**, ***$p = 0.005$, ****$p < 0.0005$. Source data
are provided as a Source data file.

We next assessed the effects of mutant YARS1 on the mobility of
synaptic vesicles at rest, a process dependent on F-actin dynamics[35].
Fluorescent recovery after photobleaching (FRAP) of Syt::eGFP-labeled
SVs (Fig. 5c), a method commonly used to evaluate the mobilization
capacity of SVs[35–37], revealed a reduced mobile fraction of SVs in
boutons of larvae over-expressing two different YARS1$^{CMT}$ proteins
compared to *YARS1$^{WT}$* larvae (Fig. 5d). We asked whether modulation of
the bundled state of the actin cytoskeleton would alleviate the deficits
in synaptic vesicle mobility. We reduced the endogenous levels of Fim
(*Fim$^{e03892}$*, Supplementary Fig. 2b), the mutant-specific genetic inter-
actor of YARS1, and this restored normal SV mobility in flies over-
expressing YARS1$^{CMT}$ alleles (Fig. 5c, d; Supplementary Fig. 9d).
Altogether, our findings suggest that YARS1$^{CMT}$ mutations affect the
organization of the presynaptic actin cytoskeleton and alter synaptic
vesicle mobility at rest in the *Drosophila* model.

Next, we assessed the effect of genetically modulating Fim levels
in the giant fiber (GF) circuit neurons in adult YARS1$^{CMT}$ flies, for
which we previously reported electrophysiological and morpholo-
gical defects[9,15]. Notably, GF interneurons have some of the longest
axons in flies, making them particularly relevant for modeling axo-
nopathies like CMT. The GF/tergotrochanteral muscle motoneuron
(GF/TTMn) synapse has decreased responsiveness in animals
expressing different YARS1$^{CMT}$ alleles (reduction of ~50% and ~80%
upon expression of YARS1$^{E196K}$ and YARS1$^{153-156delVKQV}$, respectively)
(Fig. 6a, b), associated with GF terminal morphological defects
(thinning of the GF terminal and constrictions, Fig. 6c). Remarkably,
reducing the levels of Fim (*Fim$^{Def}$*) restored the functioning of the
YARS1$^{CMT}$ flies to control levels, while it did not perturb GF function
alone or in *YARS1$^{WT}$* overexpressing animals (Fig. 6a, b). Additionally,
*YARS1$^{153-156delVKQV}$* animals with rescued GF physiology showed normal
morphology of their GF axonal terminals (Fig. 6c). Reciprocally, over-
expression of Fim (via *A307-Gal4>Fim$^{EP}$*), which exhibited no effect in
the absence of the *A307-Gal4* driver (Supplementary Fig. 10b), had
similar detrimental effect on GF electrophysiology and morphology
(Supplementary Fig. 10c) indicating these phenotypes can be a result
of actin cytoskeletal disruption. This effect was significantly
enhanced by co-expressing the YARS1$^{CMT}$ mutations (Fig. 6b). Thus,
decreasing the actin bundler Fim rescued key electrophysiological
and morphological phenotypes in the GF neurons of the *Drosophila*
YARS1$^{CMT}$ model, supporting the causal involvement of the actin
cytoskeleton in YARS1$^{CMT}$ etiology.

Neuronally expressed PLS3 had a comparable distribution pattern
(Supplementary Fig. 9e, f). In contrast, no major overlap was observed
between YARS1 and the active zone marker Bruchpilot (Brp), or the
periactive zone marker FasII (Supplementary Fig. 11b, c).

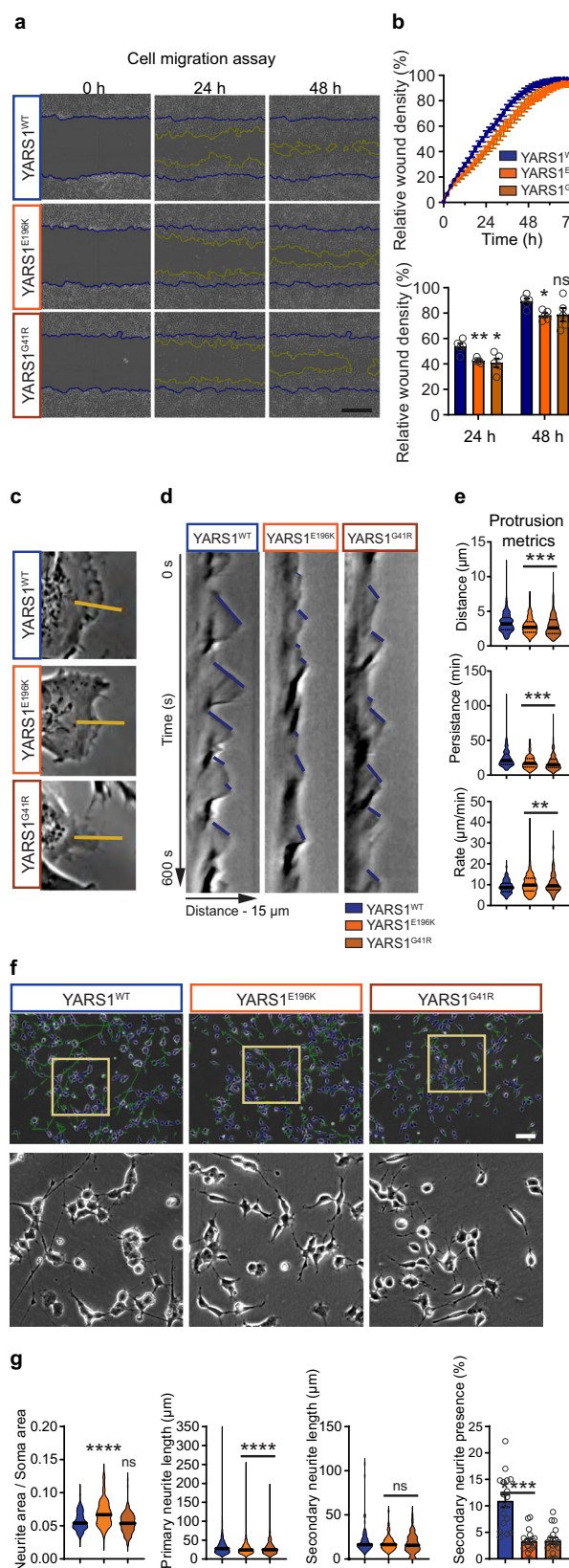

**Fig. 4 | CMT-causing YARS1 mutations disrupt cell migration, protrusion dynamics, and neurite outgrowth in SH-SY5Y cells. a** Phase-contrast time course imaging of a scratch closure after a mechanical wound induction in a confluent culture of undifferentiated SH-SY5Y cells stably expressing either YARS1[WT], YARS1[E196K], or YARS1[G41R]. Yellow lines indicate the migration front at the specific time point while blue lines show the initial position of the wound. Scale bar – 400 μm. **b** Quantifications of the cell density at the scratch over the time course of 80 h and at specific time points (24 h and 48 h); $n = 5$ independent measurements, averaged from 10-16 individual scratch wounds, taken every 2 h. Data in the bar graph in **b** presented as mean values ± SEM; at 24 h – ns – non-significant ($p > 0.9999$), *$p = 0.0317$, **$p = 0.0079$; at 48 h – ns – non-significant ($p = 0.2222$), *$p = 0.0159$. **c** Phase-contrast image sequences (10 fps for 10 min) of individual lamellipodia used to generate kymographs from 15 μm line segments (yellow lines) perpendicular to the lamellipodia movement direction. **d** Representative kymographs from YARS1[WT] and YARS1[CMT] individual lamellipodia, used to determine the protrusion displacement (micrometers on the $x$ axis), protrusion persistence (time in seconds on the $y$ axis), and the protrusion rate (displacement to persistence ratio). **e** Graph representing the protrusion metrics calculated from the kymographs. Violin plots represent at least 185 individual protrusions per genotype. After a two-sided Mann-Whitney $U$ test for wound distance ***$p = 0.0004$ and $p = 0.0002$ for YARS1[E196K] and YARS1[G41R] compared to YARS1[WT], respectively; for wound persistence ****$p < 0.0001$; for wound rate – ns – non-significant (p = 0.7920), **$p = 0.0077$ and $p = 0.0028$ for YARS1[E196K] and YARS1[G41R] compared to YARS1[WT], respectively. **f** Representative phase-contrast images of differentiated SH-SY5Y cells stably expressing YARS1[WT], YARS1[E196K] or YARS1[G41R], on which neurite outgrowth and branching were analyzed; blue traces delineate soma, green traces mark projections (primary neurites and their secondary branches). **g** Quantifications of projections' metrics in YARS1[WT], YARS1[E196K] and YARS1[G41R] cells. For neurite/soma area ratio, $n = 144$ random image fields for each genotype. Primary neurite length determined on $n = 2342$ (YARS1[WT]), 1944 (YARS1[E196K]), and 1723 (YARS1[G41R]) individual cells from at least three independent differentiations. Secondary neurite length, $n = 65$ individual cells for each genotype. Percentage of secondary neurite presence (presented as mean values ± SEM), $n = 18$ random image fields for each genotype. Median and quartiles on the violin plots are indicated by thick and thin black lines, respectively. ns – non-significant, ****$p < 0.0001$; after a two-sided Mann-Whitney $U$ test. Scale bar – 100 μm. Source data are provided as a Source data file.

disorder characterized by hypomyelination, brain stem and spinal cord abnormalities, and leg spasticity[38]. In high-speed co-sedimentation assays, we found that all three wild-type synthetases co-pelleted with preassembled F-actin filaments, indicating an ability to directly interact with F-actin (Supplementary Fig. 5i, j). Furthermore, we took advantage of our *Drosophila* model of GARS1[CMT] neuropathy, where we previously demonstrated morphological, developmental, behavioral, and electrophysiological deficits highly comparable to the YARS1[CMT] model, including morphological and functional GF defects[39]. Similarly to the YARS1[CMT] flies, we were able to alleviate the GF neuronal phenotypes upon downregulation of *Fim* (Fig. 6d, e). Thus, modulating the actin cytoskeleton is a promising rescue strategy in disease models of two CMT-causing synthetases.

## Discussion

Aminoacyl-tRNA synthetases are exemplary proteins expanding their functions over evolution through appended domains[7]. Without affecting the aminoacylation, these structural editions allow for functional divergence, thereby facilitating the necessary molecular complexity of higher organisms. By studying YARS1-associated CMT, we unexpectedly unveiled a function of wild-type YARS1 in F-actin binding and bundling, compatible with its canonical aminoacylation function. In this way, YARS1 is a new member of a family of over 200 known actin-binding proteins in humans that dynamically regulate the assembly and disassembly of F-actin, and its three-dimensional organization. Many of these ABPs compete for binding sites on actin filaments, and influence specific steps in the formation, higher-order organization, and turnover of actin networks in a concentration-, localization- and time-dependent manner[40]. Actin bundling proteins

## F-actin binding is a shared feature of aminoacyl-tRNA synthetases

Encouraged by our findings on YARS1, we tested two additional CMT-causing synthetases, GARS1 and the histidyl-tRNA synthetase (HARS1), for direct interaction with F-actin. Further, we tested aspartyl-tRNA synthetase (DARS1), which is associated with a recessive neurological

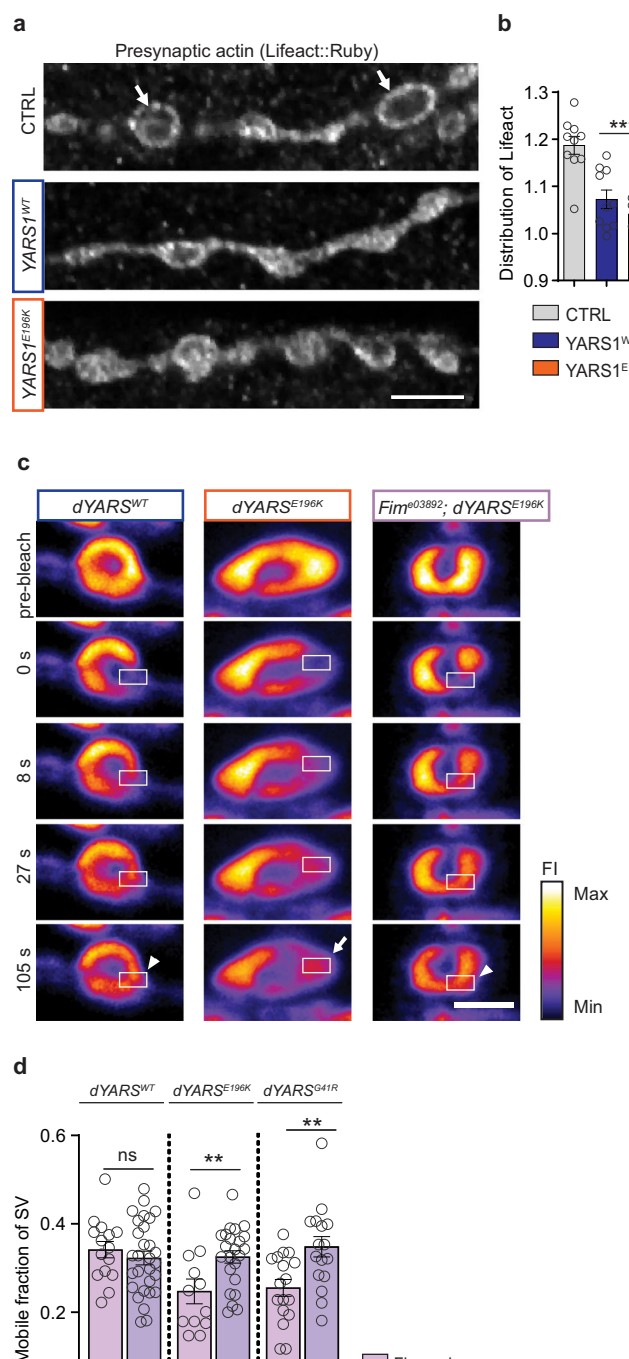

**Fig. 5 | Characterization of YARS1^CMT-induced defects at the *Drosophila* larval NMJ by genetic modulation of the actin-bundling protein Fim. a** Neuronal expression of YARS1 rearranges presynaptic F-actin (Lifeact::Ruby) from the bouton border proximity (arrows) inwards. Scale bar – 5 μm. **b** Quantification of the Lifeact::Ruby signal redistribution, as detailed in the Methods and Supplementary Fig. 8a; $n = 12$ NMJs from six larvae per genotype; ***$p = 0.0006$ and $p < 0.0001$ for YARS1^WT and YARS1^E196K respectively after a two-sided unpaired $t$ test. **c** FRAP image sequences of the SV marker Syt::eGFP co-expressed with dYARS1^WT or dYARS1^CMT. Scale bar – 2 μm. **d** The mobile fraction of synaptic vesicles, determined by FRAP of the Syt::eGFP signal, declines in dYARS1^E196K compared to YARS1^WT-expressing larval NMJs, and is restored in mutants with decreased Fim levels (*Fim^e03892*); $n = 15, 29, 12, 23, 17$ and $17$ individual boutons (from left to right). Data in bar graphs are mean values ± SEM.. **$p = 0.0064$ and $p = 0.0046$ for dYARS1^E196K and dYARS1^G41R, respectively after a two-sided unpaired $t$ test. Source data are provided as a Source data file.

crosslink actin filaments to form different types of higher-order structures and endow them with specific mechanical properties. Typically, established actin crosslinkers operate either as dimers (e.g., α-actinin) or via tandem actin-binding domains (e.g., plastins), to bring individual actin filaments together. Interestingly, YARS1 aminoacylates tRNA as a dimeric holoenzyme[12] and thus dimerization of the synthetase may underlie its bundling activity. In fact, dimerization is an enriched property for CMT-linked AARSs[2]. Notably, many actin-bundling proteins have affinity for binding actin filaments in the 1–10 μM range[16], exchanging on and off filaments on a subsecond timescale, which gives the actin cytoskeleton the plasticity for rapid reconfigurations and allows dynamic turnover of actin subunits within the individual filaments comprising the networks. Our findings suggest that the concentration at which YARS1 binds to F-actin is in the range of known established bundlers and in this way the synthetase could compete with other ABPs, some of which may be the genetic and physical interactors we identified in this study. This additional intrinsic property of YARS1 may allow the enzyme to modulate local actin networks in specific compartments of the cell, as we demonstrate by F-actin rearrangements at the synapse, and YARS1 co-localization with F-actin in HeLa cells and patient's fibroblasts.

The YARS1^E196K mutant exhibited stronger actin bundling in vitro and perturbed actin cytoskeleton organization in patient-derived fibroblasts. Both YARS1^E196K and YARS1^G41R also disturbed the actin-based global migration and protrusion formation in SH-SY5Y-derived neuron-like cells. These findings support our hypothesis that CMT neuropathy is triggered by a gain-of-function mechanism and that the actin cytoskeleton is a direct target of YARS1^CMT dysfunction. Notably, all YARS1^CMT proteins modeled so far share 3D-conformational opening that exposes sequences buried within the YARS1^WT, thereby enhancing known YARS1^WT interactions[9,12]. While it remains to be tested experimentally, it is plausible that these structural perturbations lead to the hypermorphic activity we ascribe to YARS1^CMT mutants in vitro and in vivo.

To better understand the relevance of the impaired actin-related function of YARS1^CMT for neuronal pathology of CMT, we turned to our in vivo disease model. Actin has multiple established functions in neurons, related to axonal pathfinding and synapse maintenance (synaptic morphogenesis, endocytosis, vesicle mobilization, etc) and any of these could lead to neuronal dysfunction in the long-lived neurons. As a proof of concept, we focused on synaptic vesicle mobilization, a process strongly dependent on the actin cytoskeleton[35–37]. We found this process to be impaired by the YARS1^CMT mutations at the *Drosophila* NMJ associated with local perturbations in the actin cytoskeleton. In line with an actin bundling hypermorphic function, reducing the levels of another potent bundler like Fim restored synaptic vesicle mobility in larval NMJs, and morphological and electrophysiological defects in the adult fly GF axons in YARS1^CMT mutants. Conversely, increasing the levels of Fim induced similar functional and morphological impairments of the GFs to the ones induced by the YARS1^CMT mutants. Overall, our work suggests an actin cytoskeleton-related component in the YARS1-induced CMT pathogenesis.

Besides at synapses and axons, YARS1^CMT-associated actin cytoskeleton perturbations might potentially affect molecular processes in other subcellular compartments. The expected pleiotropic effect is in line with the existing mechanistic hypotheses about YARS1^CMT pathogenesis raised by us or others. For example, we demonstrated that upon stress conditions a fraction of YARS1^WT enters the nucleus and acts as a powerful transcriptional regulator of over 700 genes, while the CMT mutations dysregulate this function. Because of the global nature of the transcriptional effect provoked by YARS1^WT or YARS1^CMT, it is unlikely that an interaction with a single transcription factor will be involved. This is corroborated by our findings that modulating the known YARS1 interaction with the E2F1 transcription factor was

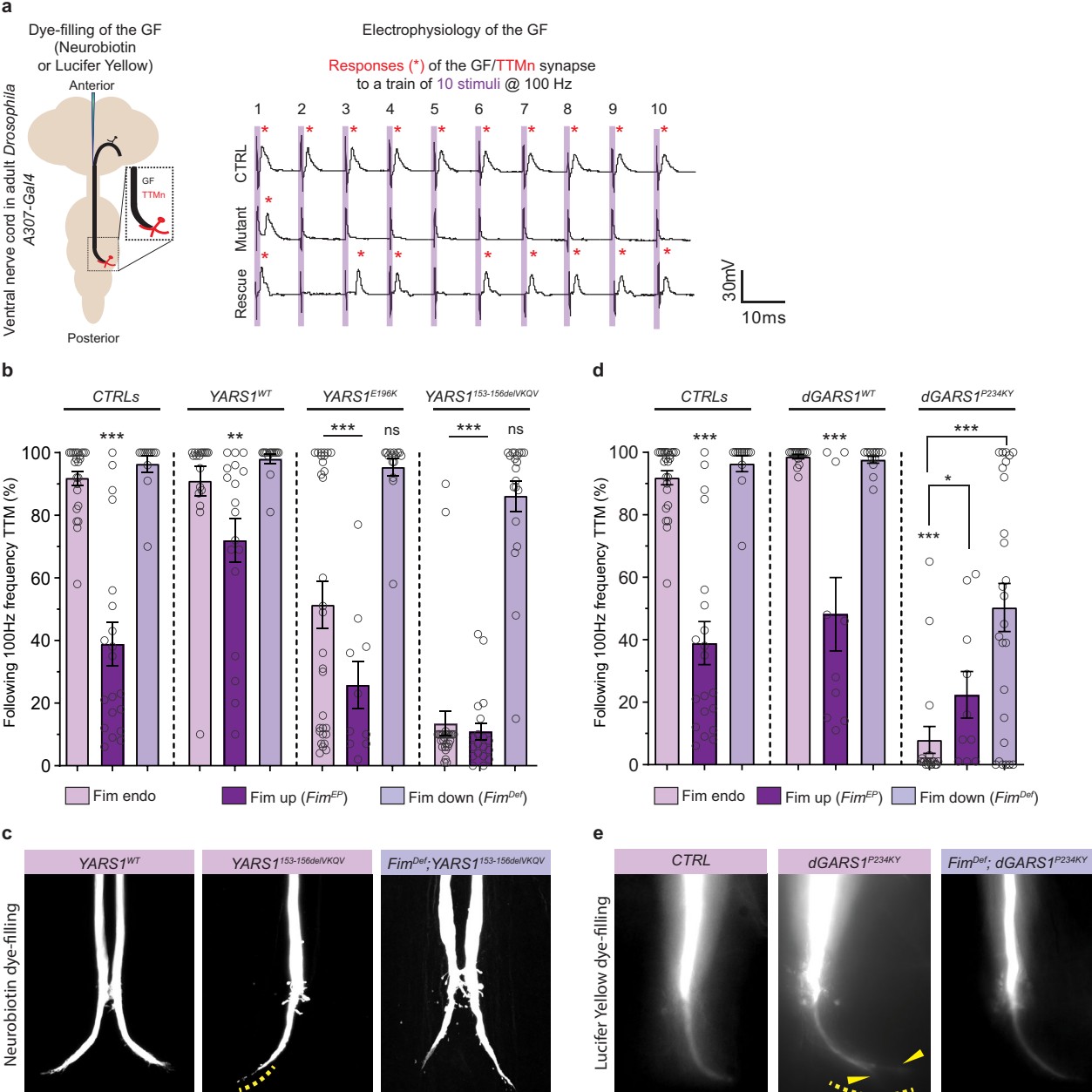

**Fig. 6 | *Fim* alters neuronal phenotypes in the giant fibers in YARS1 and GARS1 neuropathy *Drosophila* models. a** Evaluation of the giant fiber-tergotrochanteral motoneuron connection (GF/TTMn; other connections omitted for simplicity) in the ventral nerve cord of adult flies. Sample traces of a train stimulation and GF/TTMn recordings in controls, severely impaired YARS1CMT mutant (*YARS1^153-156delVKQV^-*expressing) and animals with improved GF/TTMn function in *Fim^Def^* background. **b** Quantification of the ability of the GF/TTMn synapse to follow repetitive stimuli in control, YARS1WT and CMT mutant animals, in the background of endogenous-, increased- or decreased Fim expression; $n = 24, 20, 12, 19, 18, 14, 30, 10, 15, 28, 19,$ and 20 (from left to right) giant fiber recordings from eight days-old female flies. **$p < 0.01$ and ***$p < 0.001$, one-way ANOVA with Bonferonni Multiple Comparison Test. **c** GF terminal morphology visualized by neurobiotin dye injection in *YARS1^WT^*, *YARS1^153-156delVKQV^*, and flies with improved GF/TTMn responses

(*Fim^Def^; YARS1^153-156delVKQV^*). Yellow dashed line depicts constrictions at the axonal terminal in the mutant. Scale bar – 20 μm. **d** The same electrophysiological paradigm was used to evaluate the ability of the GF/TTMn synapse to follow repetitive stimuli in control, *GARS^WT^* and *GARS^CMT^*– expressing animals, by changing the expression levels of Fim; $n = 24, 20, 12, 14, 10, 12, 18, 10,$ and 26 (from left to right) giant fiber recordings from 8-day-old female flies; *$p = 0.0159$ and ***$p < 0.001$, by a two-sided Mann-Whitney test. Controls are reused in graph **b** and **d**. Data in **b** and **d** presented as mean values ± SEM. **e** GF terminal morphology visualized with lucifer yellow GF dye-filing in *CTRL*, *GARS1^CMT^*, and flies with improved GF/TTMn responses (*Fim^Def^; GARS1^CMT^*). Yellow dashed line depicts the thin, irregular axonal terminal that splits in two (yellow arrowheads) in the mutant. Scale bar – 50 μm. Source data are provided as a Source data file.

insufficient to alleviate the neurodegeneration in the *Drosophila* model. Rather, the genetic or pharmacological exclusion of YARS1CMT from the nucleus rescued the hallmark features of the pathology[9]. In light of the actin cytoskeletal perturbations we established in different neuronal and non-neuronal YARS1CMT models, it is plausible that, in part, the transcriptional changes are a downstream effect of actin

cytoskeleton remodeling in the nucleus. Indeed, nuclear actin has been linked to a variety of processes including transcription and transcription regulation, RNA processing and export, dynamic chromatin organization and remodeling, DNA repair, or nuclear envelope assembly[41]. To this end, numerous actin-remodeling proteins are localized there or translocate upon specific stress stimuli to

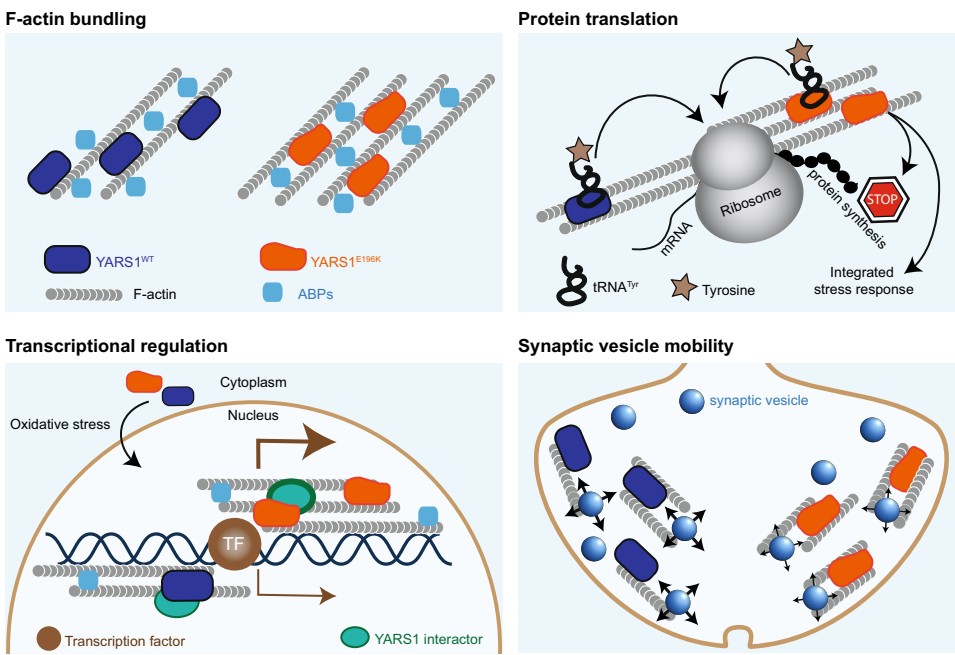

**Fig. 7 | Illustration of the known cellular processes implicated in YARS1$^{CMT}$.** The enhanced F-actin bundling properties described for YARS1$^{CMT}$ in this study might contribute to global protein synthesis inhibition, activation of the integrated stress response, transcriptional disregulation, and impaired synaptic vesicle mobility.

orchestrate the dynamic equilibrium of nuclear actin[41]. YARS1 might influence the state of nuclear actin and related processes in different ways. Via its actin-binding and bundling function, it can potentially directly modify the organization of the nuclear microfilaments, or it can do this via interactions with other actin-binding proteins. Future developments of more sensitive molecular tools for visualization of nuclear actin that preserve its versatile organization, functions, and binding partners will enable us to test these hypotheses in the context of CMT.

Studies in *Drosophila* YARS1$^{CMT}$ and GARS1$^{CMT}$ models established inhibition of global protein translation in fly neurons as a common feature. For GARS1$^{CMT}$ flies, the translational arrest was not due to loss of tRNA$^{Gly}$ aminoacylation and was not alleviated by over-expression of the *Drosophila* wild-type GARS1 enzyme, suggesting a gain-of-function mechanism[42]. Furthermore, recent findings in *Gars1*$^{CMT}$ and *Yars1*$^{CMT}$ mice models revealed neuronal upregulation of the ATF4-dependent integrated stress response (ISR) pathway as another commonality between both CMT-related synthetases[43]. Sequestration of tRNA$^{Gly}$ by GARS$^{CMT}$ was demonstrated to cause ribosome stalling, translational arrest, and upregulation of the ISR via the GCN2 kinase. Intriguingly, overexpression of tRNA$^{Gly}$ alleviated (but not rescued) the CMT-specific phenotypes in the fly model, while genetic or pharmacological inhibition of GCN2 markedly reduced the progression of the neuro-degenerative phenotypes in the *Gars1*$^{CMT}$ mice models tested[43,44]. In contrast, the mechanisms behind the global translation arrest (fly) and the upregulated ISR (mouse) in YARS1$^{CMT}$ models remain unknown. The findings of this study are in line with the triggered cellular responses in the YARS1$^{CMT}$ disease models. The translational machinery is intrinsically connected to the actin cytoskeleton[45]: F-actin is the main cytoskeletal component that organizes spatially and temporally the translational apparatus[46] and regulates both translation and ISR[47], while perturbations of the actin network lead to translational arrest. Also, actin dynamics tunes the ISR via eIF2a, the upstream regulator of ATF4[47]. Conversely, components of the translational machinery including polyribosomes, initiation, and elongation factors can contribute to actin remodeling via directly binding to actin[48-50]. For example, eEF1A, important for the delivery of aminoacylated tRNA to the polyribosomes, binds to and bundles F-actin independent of its

role in translation[51]. eEF1A uses its actin remodeling function to switch off/on the global translation in a GCN2-dependent manner, and in this way establishes a functional link between protein synthesis, ISR, and actin cytoskeleton[52]. Our findings that YARS1 performs an actin-organizing function similar to eEF1A establishes a link between the global translational arrest and the observed ISR in the *Drosophila* and mouse YARS1$^{CMT}$ models, respectively. Altogether, it is plausible that the YARS1$^{CMT}$ dysfunction is directly associated with the actin remodeling we describe in our study, with pleiotropic effects on neuronal transcription[9], translation[42], and adaptation to stress[43], collectively inducing progressive peripheral neuropathy as CMT (Fig. 7).

The findings for YARS1$^{CMT}$ are also relevant for GARS1$^{CMT}$ and potentially other forms of CMT associated with defects in aminoacyl-tRNA synthetases. We provide evidence of direct binding between F-actin and glycyl-, histidyl- and aspartyl-tRNA synthetases, while the methionyl-, phenylalanyl-, and seryl-tRNA synthetases were found in complexes with actin in mammalian cells[48-50]. Hence, five out of six CMT-causing synthetases are associated with the actin cytoskeleton so far. While we demonstrate that YARS1 is an effective modulator of actin cytoskeleton, the impact on F-actin organization for the others remains to be delineated. In addition, we demonstrate a beneficial effect of manipulating the actin cytoskeleton on the neurotoxicity in the GARS1$^{CMT}$ *Drosophila* model, suggesting that, in part, the observed global translational arrest and ISR response in the flies might be mediated by the actin cytoskeleton. In this way, we suggest a potential common mechanistic denominator among the AARSs-related CMT neuropathies. Our data complement previously published findings about mechanistic and structural commonalities and point out that a complex interdependence of the concentration of charged tRNA, actin organization, ISR, and the AARS' subcellular localization might be at the basis of the synthetase-induced CMT (Fig. 7). This interdependence should be tested experimentally in the future, preferably in a systematic comparative study using unified disease models. The potential discovery of a shared neurotoxic signaling pathway(s) might facilitate the development of drugs for a greater number of individuals afflicted with very similar symptoms.

Increasing evidence implicates actin-binding and -regulatory proteins in the degeneration of neurons with long axons, e.g., central

and peripheral motor and sensory neurons. These are the most polarized cells in our body, and they are long-lived and particularly vulnerable to changes in their cytoskeletons over time. Our findings provide further evidence supporting this causal link and suggest that maintaining the dynamic equilibrium of actin cytoskeleton organization is crucial for the lifelong support of neuronal integrity and function.

## Methods

### Ethical approval

We have complied with all relevant ethical regulations for animal and human testing and research. The study was approved by the Ethical committee of the University of Antwerp and University Hospital Antwerp (14/15/188). Informed consent was obtained from all individuals included in the study.

### Drosophila genetics

All *Drosophila* crosses were performed at 25 °C, 12 h light/dark cycle, on a standard NutriFly medium (Flystuff). *UAS-YARS1* flies, expressing full-length human (YARS1) and *Drosophila* (dYARS1) proteins, were previously described in ref. [15]. To generate *UAS-YARS1::GFP* transgenic flies, eGFP was inserted at the N-terminal of human YARS1$^{WT}$ or YARS1$^{E196K}$ in the pEGFPC1 vector, which was sub-cloned into the pUAST transformation vector. *UAS-dGARS1* flies carrying the cytoplasmic isoform of the *Drosophila* orthologue of glycyl-tRNA synthetase (wild type and mutants) were previously reported in ref. [39]. All constructs were sequence verified and transgenic flies were generated using standard procedures. For each construct, multiple transgenic lines were established. All transgenic flies used are in the w$^{1118}$ genetic background.

### Enhancer promoter lines used for the retinal screen

The *P(Mae-UAS.6.11)*[23], *P(BacWH)*[24], *P(EP)*[2522], and *P(EPg)*[25] enhancer-promoter lines on the X-chromosome were obtained from the Bloomington *Drosophila* stock center (BDSC). The following *EP* lines, inserted in the direction of the *Fim* gene expression were screened for eye phenotypes: *P(XP)Fim$^{d02114}$*, *P(XP)Fim$^{d05016}$*, *P(XP)Fim$^{d03334}$*, *P(EP)Fim$^{G10929}$*. In addition, we used the following BDSC lines: the Fim protein trap line *P(PTT-GC)Fim$^{CC01493}$*, *UAS-Syt::eGFP* (*BDSC_6926*), *UAS-Lifeact::Ruby* (*BDSC_35545*), *UAS-Act5C::GFP* (*BDSC_9258*), *UAS-mCD8::RFP* (*BDSC_32218*), *UAS-Pod1* (*BDSC_8800*), *P(EPgy2)coro$^{EY05114}$* (*BDSC_19703*), deficiency line covering *Fim* (*Fim$^{Def}$*) is *BDSC_4741*, and the *Fim$^{XP}$* (*BDSC_19171*) and *Fim$^{e03892}$* stock was obtained from the Exelixis collection. *UAS-RNAi* lines, including the one used for *IKKε* (*VDRC_103748*) were from the Vienna Drosophila Research Center. The following Gal4 lines were used: *GMR-Gal4*, *nSyb-Gal4*, *A307-Gal4*. For the FRAP assay, *nSyb-Gal4* was recombined with *UAS-Syt::eGFP* on the third chromosome. The insertion sites of the *EPs* were determined by inverse PCR following the protocol indicated in the Gene Disruption Project (http://flypush.imgen.bcm.tmc.edu/pscreen/). Unless otherwise stated, we assessed female flies throughout the experiments.

### Retinal degeneration screen

We chose to screen for *EP* lines on the *Drosophila* X-chromosome as a strategy to identify YARS1$^{CMT}$-relevant genetic interactions in an unbiased manner, and yet avoid a lengthy whole-genome screening, or screening for *EP* lines of the much larger chromosomes 2 or 3. The screening was performed as described in ref. [39]. Briefly, *GMR-Gal4; UAS-YARS1$^{E196K}$/TM6B* virgins were crossed with *EP* males. In case of *EP* male sterility or lethal *EP*-element insertion, crosses were repeated vice versa, using *EP* virgins and *GMR-Gal4; UAS-YARS1$^{E196K}$/TM6B* males. In F1, at least 20 female flies heterozygous for *GMR-Gal>YARS1$^{E196K}$>EP* and *GMR-Gal4>TM6B>EP* genotypes were compared with each other. Initial positive hits were selected when rough eye phenotype was present in flies with the first genotype and was absent in flies with the

second genotype, respectively. The crosses for the positive hits were repeated at least three independent times. In a next step, the initially positive hit went through a rigorous validation process, where the major criterium to be claimed as a genuine CMT modifier was to induce a rough-eye phenotype only upon co-expression with *YARS1$^{E196K}$* and not with *YARS1$^{WT}$*. Additionally, the genetic interaction had to be evolutionary conserved (i.e., present both with human and *Drosophila* versions of *YARS1$^{E196K}$*) (see Fig. 1a for a schematic representation of the screening strategy).

### Generation of *UAS-Fim, UAS-PLS2, UAS-PLS3* transgenic flies

Full-length *Drosophila* Fim cDNA (clone LD05347), covering the longest ORF and over-expressing all three Fim isoforms (A, C, and D), was obtained from the BDGP *Drosophila* Gold Collection, while PLS3 and PLS2 pCDNA3 constructs carrying the responding full length human cDNAs were kind gift from Dr. J. Gettemans (UGent). Using the Gateway® system (Life Technologies), cDNA from the three constructs was sub-cloned into the pUASg.attB expression vector. The ΦC31 system was used to generate independent transgenic lines at landing sites attP40 and attP2 (BestGene).

### Immunoprecipitation and mass spectrometry in HEK293 cells

C-terminally FLAG-tagged human YARS1$^{WT}$ was generated by transferring the FLAG-TEV-ProteinA downstream of the YARS1-containing Gateway® cassette to the Flp-in vector pCDNA5/FRT, via Gateway® recombination (Thermo Fisher Scientific). To identify the binding partners of YARS1, HEK293 cells (purchased from DSMZ, ACC 635) were transiently transfected with the FLAG-YARS1 vector. The immunoprecipitation experiment was performed in triplicates. The cells were collected after 48 h of transfection and lysed in lysis buffer (50 mM Tris-HCL at pH 7.4, 100 mM NaCl, 2 mM MgCl$_2$, 1% NP40, protease inhibitor). The residual cell debris was removed using 20 min high-speed centrifugation (13,000 rpm). The cell lysate was incubated for 1 h with 50 μl Flag beads (Miltenyi Biotec) on a rotary shaker at 4 °C. The resin was washed several times with a low salt-containing buffer (50 mM Tris-HCL pH 7.4, 100 mM NaCl) to remove the residual NP40 detergent. For identification of YARS1 binding partners, on beads trypsin digestion was performed like in ref. [53]. The fraction containing the bound protein was reduced, alkylated, and digested overnight with trypsin (1/100 trypsin/protein ratio). At the end, the peptide mixture was acidified using formic acid. The mixture was cleaned using stage tip protocol and resulting peptides were analyzed using a Q Exactive Orbitrap mass spectrometer (Thermo Fisher). The mass spectrometry and data analysis were performed as described in ref. [54]. Briefly, the quantitative proteomics software package MaxQuant version 1.5.3.8[55] was used to obtain label-free quantification (LFQ) from the mass-spec data. The LFQ value for each protein was calculated with the MaxLFQ algorithm within 2 min retention time window using both unique and razor peptides.

### Gene ontology analyses

For the genetic screen performed in the *Drosophila* CMT model, the genes potentially targeted by an *EP* insertion were manually curated for each EP line used in the retinal degeneration screen, either from the genotype description of the *EP* itself, or by a Flybase search[56]. The generated list does not exclude other potentially affected genes. The EP-targeted genes' annotations are based on the Flybase genome release FB2020_06. We used the PANTHER Classification System, version 16.0 to retrieve gene ontology (GO) information[57]. The functional classification of the EP-targeted fly genes and more specifically, their assignment to different protein classes was explored.

### Bioinformatic analysis

Expression data and functional annotations were retrieved from Fly-Base and modENCODE. ClustalOmega was used for protein sequence

alignment[58]. DroID data base was used to explore 23 predicted protein interactions in the targeted screen for genetic modifiers (See Supplementary Fig. 1e)[26].

### Characterization of *Drosophila* transcript levels

The transcript levels of *Fim* in *nSyb-Gal4 > UAS-Fim, nSyb-Gal4 > Fim*^EP^ as well as *Fim*^e03892^ and *Fim*^Def^ fly lines, were determined upon qPCR of cDNAs from fly head extracts using standard procedures, from 2 to 3 independent experiments, each performed in technical triplicates. The following primers were used: 5′-ACGCACAGCAAACCGGTCGA-3′ and 5′-CACCCATACACACTCGTTGCTGGT-3′.

### Scanning electron microscopy

Adult flies were terminally anesthetized with ether and directly mounted on aluminum stubs (Electron Microscopy Sciences) without any tissue processing steps. Immediately after gold sputter coating, the eyes of the flies were imaged using scanning electron microscopy (SEM) with an SEM505 microscope (Philips).

### *Drosophila* larval immunohistochemistry and microscopy

Third instar larvae were dissected in HL3 buffer and subsequently fixed in HL3 + 3.7%PFA for 20 min. Tissue was permeabilized using 1× PBS with 0.2% Triton-X and 5% BSA. Staining was performed using the following probes/antibodies: horseradish peroxidase (HRP; Jackson ImmunoResearch Laboratories, 1:250), Disc Large 1 (4F3, DHSB, 1:50), mouse monoclonal YARS1 (Abnova, 1:500), rabbit polyclonal GFP (Invitrogen, 1:2000), rabbit polyclonal RFP (Abcam, 1:250), mouse monoclonal Brp (nc82, DHSB, 1:100), mouse monoclonal FasII (1D4, DHSB, 1:250), mouse monoclonal Synapsin (SynORF1, 3C11, DHSB, 1:500). Alexa Fluor®−488 and Alexa Fluor®−546 secondary antibodies were used (Invitrogen, 1:1000). Muscle 6/7 of abdominal hemisegments 3 and 4 were imaged.

Laser scanning confocal microscopy was performed on a Carl Zeiss LSM700 microscope equipped with a 20× Plan-Apochromat (0.8 NA) or 63× Plan-Apochromat (1.4 NA) objective. Super-resolution structured illumination microscopy was performed on a Zeiss ELYRA S.1 microscope equipped with a 63× Plan-Apochromat objective (1.4 NA). For a description of methods used to calculate Lifeact-RFP distribution at synaptic boutons, please see Supplementary Fig. 7.

### Assessment of Act5C::GFP at boutons

Female virgin flies with the genotype *nSyb-Gal4>UAS-Act5C::GFP* were crossed to *UAS-YARS1*, *UAS-PLS3*, or *UAS-CD8::RFP* male flies. Third instar larvae were dissected in HL3.1, fixed in 4% PFA solution (in HL3.1) for 10 min, washed three times for 10 min with 1XPBS, blocked with 2% BSA/Normal Goat Serum, 0.1% Triton-X 1X PBX, incubated with α-GFP nanobodies (Nanotag Biotechnologies) to amplify the Act5C::GFP signal and HRP-Rhodamin red to label the neuronal membrane (both at 1:250 in blocking solution) for 2 h at room temperature, washed three times for 10 min in 0.1% PBX and mounted on glass microscopic slide in Diamond ProLong. NMJs on muscle 6/7 on both sides of abdominal segment A3 and A4 were imaged using a Nikon Ni-E upright microscope equipped with a Yokogawa CSU-W1 spinning-disk head, an Andor iXon 897U EMCCD camera and Nikon Elements AR software. A 60× (NA 1.4) oil immersion objective was used to image the NMJs. Image acquisition settings were identical for all images. Act5C::GFP-labeled actin assemblies were assessed on maximum-intensity projection images. To analyze actin in boutons, boutons were cropped, and actin assemblies were analyzed with the pipeline described below. The Trainable Weka Segmentation (TWS) machine-learning tool[59] in Fiji was used to manually annotate GFP-positive actin assemblies with different fluorescence intensities, and to train a classifier that will automatically segment these structures. The segmented objects were subjected to Huang auto thresholding to obtain binary masks. Next, we applied a Watershed processing on the binary image, to improve the

isolation of individual neighboring objects from the diffraction-limited images. We performed particle analysis on the segmented actin assemblies to obtain their number and area. Mean fluorescence intensity of individual actin assemblies was measured by applying the mask on the original image. The number of actin assemblies was normalized to the bouton area. To determine the bouton area using TWS, we developed different classifier by annotating the Act5C-positive bouton (Gaussian blur 2).

### Fluorescent recovery after photobleaching assay at the NMJ

Third instar larvae expressing eGFP-tagged synaptotagmin were dissected in HL3 on a Sylgard polymer block and placed in a 50 mm glass bottom imaging dish (WillCo Wells B.V.). FRAP time series were acquired on an inverted Zeiss LSM700 confocal laser scanning microscope using a 63× Plan-Apochromat objective (1.4 NA) and controlled by Zen2009 software. A 488 nm diode laser was used to image and photobleach the eGFP signal. Pre- and postbleach images (16 bit, 512 × 512 pixels, 0.033 µm/pixel) were acquired every 3.8 s using low laser power and closed pinhole (1.0 Airy units). For photobleaching, a rectangular region of interest (ROI; 40 × 20 pixels) covering part of a large (>3 µm) bouton was drawn. Full laser transmission (100%) was used to bleach the ROI during 0.3 s. The total duration of time series was 1 min 53 s and consisted of two prebleach frames and 28 frames after the photobleach. To analyze the FRAP data, a purpose-built ImageJ[60] macro was used to process all FRAP series in batch. In brief, Zeiss lsm-files were imported using the Bioformats Java library of ImageJ and mean pixel gray values were measured in the bleach-ROI (imported as metadata during the lsm file import). These mean intensity values were normalized using the highest pre-bleach value and the lowest overall value, and the data of the resulting normalized post-bleach series was fitted to the typical exponential recovery curve using the ImageJ "Curve Fitting" tool (equation used: $y = a^*(1-exp(b^*x))$). The FRAP mobile fraction (parameter $a$ in the equation) was determined for every photobleached bouton. Boutons with poor curve fitting ($R^2 < 0.95$) were excluded from the data.

### Giant fiber electrophysiology

Intracellular recordings from the tergotrochanteral muscle (TTM) of adult male and female flies were obtained as previously described[9,15,61]. Briefly, giant fibers (GFs) were activated with 0.03 ms pulses of 30–60 V using two tungsten electrodes inserted into the brain (Grass S44 stimulator, Grass Instruments). Saline-filled glass electrodes were used for recordings from the TTM and a tungsten electrode in the abdomen served as a ground electrode. The recordings were amplified (Getting 5 A amplifier, Getting Instruments) and the signals were stored and analyzed using pCLAMP software (Molecular Devices). The ability of the GF to TTM pathway to follow high-frequency stimulations was assessed with ten trains of ten pulses given at 100 Hz with a 1 s interval between the trains. The average following frequencies were calculated as percent responses.

### Dye injections and immunohistochemistry of the giant fibers

Dye injection and immunohistochemistry methods have previously been described in detail[62,63]. In brief, the ventral nerve cord of adult *Drosophila* was dissected and mounted dorsal side up on VECTA-BOND™ (Vector Labs) coated 0.9–1.1 mm etched slides. An 80–100 MΩ glass electrode filled with a dye solution of 10% w/v neurobiotin (Vector Labs) and tetramethyl rhodamine-labeled dextran (Invitrogen) and backfilled with 2 M potassium acetate was used to inject the dyes into the GF axons by passing depolarizing current. Samples were fixed in 4% paraformaldehyde and were prepared for confocal microscopy as described previously[62,63]. Streptavidin-Cy2 conjugate (Jackson ImmunoResearch; 1:750) was used to visualize neurobiotin. Samples were scanned at a resolution of 1024 × 1024 pixels, 2.5× zoom, and 0.5 µm step size with a Nikon C1si Fast Spectral Confocal system using

a 60× oil immersion objective lens. Dye filling of the GFs with Lucifer Yellow was performed as described in ref. [15].

## Statistical analysis

GraphPad Prism software was used to generate graphs and to perform statistical tests throughout the manuscript. One-way ANOVA with multiple comparison test was used to analyze the data from the giant fiber system electrophysiological recordings and expression level determination in the SH-SY5Y cell lines. Two-sided unpaired t-test or Mann-Whitney $U$ test were used to analyze the rest of the data. All data are presented as mean ± standard error of mean (SEM), unless otherwise stated. Chi-square test was used to analyze data presented in graphs in Fig. 3d, e.

## Patient material

Dermal fibroblasts from one YARS1[E196K/WT] patient (male, 44 y) and one control individual (male, 42 y) were sampled using standard procedures and after obtaining their written informed consent. The study complies with the ethical guidelines of the Medical University-Sofia, Bulgaria and University of Antwerp, Belgium and was approved by the respective local institutional review boards. The biosamples are registered in the Biobank Antwerp, Antwerp, Belgium; ID: BE 71030031000.

## Fibroblasts cell culture, staining procedures, and microscopy

Fibroblast were grown on Y-micropatterned fibronectin-coated CYTOO plate (CYTOO SA) (size 700 μm²), using company-recommended protocol. Briefly, standard fibroblast cultures were maintained in DMEM/F12 medium (Thermo Fisher Scientific) supplemented with 15% FBS. For the assay, cells of confluent bottles were collected by gentle trypsinization and diluted to a concentration of ~30,000 cells/ml. 100 μl (~3000 cells) were homogeneously dispensed into each well. Cells were fixed within 6 h after adhesion, using 3.7% paraformaldehyde. Staining procedure involved 2 washes in PBS, permeabilization using 0.1% Triton X100 and staining with Alexa-Fluor™ 594 Phalloidin (Thermo Fisher Scientific) according to manufacturer's instructions. Cells were imaged using laser scanning confocal microscope Carl Zeiss LSM700 equipped with a 40× (1.3 NA) Plan-Neofluar objective. Voxel size: 0.1563 × 0.1563 × 0.9474 μm³.

## Image analysis of fibroblasts

ImageJ software[60] was used to assess the fluorescent intensity and distribution of the phalloidin signal using the following workflow: images underwent background subtraction (rolling ball radius of 10 pixels). The data sets for fibroblasts of control individuals (YARS1[WT]) and YARS1[E196K] patients contain comparable cell numbers and have comparable fluorescent intensities. Fluorescent intensity of the Phalloidin signal as raw integrated pixel density was measured in ImageJ, using the 32-bit sum intensity projection (SUM) on a segmented cell. Segmentation was performed using Huang autothresholding>create mask>fill holes, and the generated mask (mask_whole) was used to measure the fluorescent intensity on the SUM of an individual cell. This mask was eroded by 12 pixels or ~1.9 μm (pixel size ~0.156 μm) to render new mask (mask_12erode) to select the region of interest excluding the pronounced stress fibers lining the periphery of the triangular cell. The fluorescent intensity on the periphery (including these the stress fibers) was measured by applying the two different ROIs (mask_whole and mask_12erode) on the SUM, using the XOR function in the ROI manager and Measure. The data points on the graphs represent individual cells.

## Qualitative analyses of actin cytoskeleton in fibroblasts

The same data set of cells was used to score the frequency of phenotypes regarding the abundance of stress fibers, speckled actin structures as well as the number of actin-rich apices. Stress fibers were present in all cells, and we scored as prominent when they resembled the three representative images in control fibroblasts and the last image of the YARS1[E196K] in Fig. 3a. The first two representative images of YARS1[E196K] cells would be scored as having "some" stress fibers as they appear interrupted (first image) and thinner (second image). Actin speckles are the structures marked with asterisk in the second image in CMT fibroblasts, which would be scored as having "prominent" speckles, the first images in both control and CMT cells would be scored as having "some", and the second image in control cells would be scored as practically having "none". First image in control cells would be scored as having "1" actin-rich apex, first image in CMT cells as having "2", and the third image in mutant cells as having "3" actin-rich apices.

## Generation of SH-SY5Y cell lines expressing YARS1 constructs

Constructs were created using the Gateway® recombination technology (Life Technologies). The open reading frame of YARS1 was amplified by PCR using specific primers to allow insertion of the product in a pcDNA5/FRT (Invitrogen) vector. YARS construct was tagged at its C-terminal end with ProtA-TEV-Flag tags. The YARS1[CMT] mutations (p.E196K and p.G41R) were generated by site-directed mutagenesis. All constructs were validated by Sanger sequencing and transferred by recombination to a pLenti6 destination vector (Life Technologies). Stable cell lines were generated by lentiviral transduction of SH-SY5Y human neuroblastoma cells (purchased from ATCC, CRL 2266) as described in ref. [64]. SH-SY5Y cell lines were cultured at 37 °C and 5% $CO_2$ in a humidified atmosphere in DMEM/F-12 complete medium (Gibco) containing 10% fetal bovine serum (Gibco) and 1% penicillin/streptomycin (Gibco).

## Cell migration assay in SH-SY5Y cells

Cell migration was measured using the Incucyte® Cell Migration Kit (Sartorius). To this end, SH-SY5Y neuroblastoma cell lines were seeded in Incucyte®Imagelock 96-well plates at $5 \times 10^4$ cells per well. The next day, at 100% confluency, a standardized strip of cells was scraped away in each well using the Incucyte® Woundmaker Tool (Sartorius). Every well was subsequently imaged with phase contrast every hour for 80 h in the instrument. Analysis was performed using the Incucyte® Scratch Wound Analysis Software Module to segment at each time point the wound and the front of the migrating cells. The relative wound density (area of the wound normalized to the area of the initial wound at time point zero) is a measure for cell migration. Per plate at least 16 wells (=16 wounds) were imaged. Low-quality wounds (irregular, occurrence of cell debris) were removed (<10% of wells) and per plate a mean wound density was calculated per time point. Presented data is the average and SEM of five independent experiments (=96 well plates). ANOVA and Tukey HSD statistical testing was performed to compare the genotypes on 24 h and 48 h time points.

## Protrusion dynamics in SH-SY5Y cells

SH-SY5Y neuroblastoma cell lines were seeded in 35 mm diameter Ibidi μDishes (Ibidi) to a density of $4.5 \times 10^5$ cells per mL ($9 \times 10^5$ cells per dish). At full confluency and 2-to-4 h prior to imaging, a strip of cells was scraped away with a pipet tip to obtain a front of migrating cells. Phase contrast imaging was done on a Zeiss Axiovert 200 M microscope equipped with a Zeiss AxioCam MR3 camera and a LCI Plan-Neofluar 63×/1.30 Imm Korr Ph 3 objective. Images were captured every second for 10 min, resulting in 601 time frames (16-bit, 1388 × 1040 pixels, 102 nm pixel dimension). Parameters describing the dynamics of cellular protrusions were measured using the Fiji distribution of ImageJ, similar to the analysis performed in[65,66]. To this end, straight line segments (15 μm long) were drawn perpendicular to the movement direction at the front of active lamellipodial protrusions. From these lines, kymograph images were generated ("Multi Kymograph" command, linewidth = 3) and saved. On the kymograph images, line segments were drawn covering entire individual

protrusions that move forward at a constant speed (representing lamellipodia, straight lines on kymographs). The displacement in the horizontal (X) direction is the protrusion distance (μm) while the displacement in the vertical (Y) direction is the protrusion persistence (in seconds). The protrusion rate (μm/s) can be calculated by taking the ratio of both and is also represented by the angle of the line segment. Per genotype, we acquired at least ten image sequences, resulting in 21–49 kymographs and at least 185 measured individual protrusions. ANOVA and Tukey HSD statistical testing was performed to compare the genotypes.

### Differentiation and neurite outgrowth of SH-SY5Y cells

SH-SY5Y cells were seeded 24 h prior to differentiation in 24-well plate to a density of $1 \times 10^3$ cells per well. Cells were differentiated in B-27™ Plus neuronal culture system (Life Technologies) supplemented with 20 μM retinoic acid (Merck Life Science), 1% L-Glutamine (Gibco), and 1% penicillin/streptomycin (Gibco). Medium was refreshed every other day. Phase-contrast imaging was done on a Zeiss Axiovert 200 M microscope equipped with a Zeiss AxioCam MR3 camera and 20× phase contrast objective. Three images per well were captured at day 8 of differentiation. To extract the total area covered with neurites and soma in each image, we used a custom-developed ImageJ script to automatically segment both neurites and soma, using combinations of simple image operations that can i) remove noise (noise reduction filters), ii) separate fine from coarse structures (rolling ball algorithm or Fast Fourier Transform), iii) separate bright from dark regions (automatic intensity thresholding) and iv) exclude segmented regions based on size or shape (morphological operations). The resulting segmentation masks were used to calculate the ratio of skeletonized neurites per cell body area. In addition, we manually traced individual neurite structures. To this end, images were converted to 8-bit and analyzed with NeuronJ plugin in ImageJ, a commonly used tool for semiautomatic tracings and measurements of neurites[67]. Any projection from SH-SY5Y cell body was considered a "primary neurite", whereas projections branching from primary neurites were considered a "secondary neurite". Three wells per genotype and three images per well were analyzed, and data from two experiments (repetition) was pooled, resulting in over 600 tracings in total. The overall distribution of primary and secondary neurites lengths was plotted. For each image, the fraction of secondary neurites (over the total) was also calculated. One-Way Anova was used for statistical analysis.

### YARS1-eGFP and LifeAct-mCherry microscopy in HeLa cells

HeLa cells were incubated at 37 °C and 5% $CO_2$ in a humidified atmosphere and maintained in DMEM complete medium (Gibco) containing 10% fetal bovine serum (Gibco), 1% L-Glutamine (Gibco) and 1% penicillin/streptomycin (Gibco). Cells were transiently transfected using Lipofectamine™ 2000 transfection reagent (ThermoFisher Scientific). Briefly, cells were seeded in a six-well plate the day prior to transfection in complete medium. At 70–80% confluency, cells were co-transfected with 1 μg of the mCherry-Lifeact-7 plasmid (Addgene) together with wild-type or mutant pEGFP_YARS1 plasmid (1:1 ratio). The next day, cells were reseeded in 35 mm glass bottom imaging dishes (MatTek Corporation). Fixation (15 min 4% PFA, Laborimpex) or live cell imaging was performed the next day (48 h after transfection) on a Zeiss LSM700 laser scanning confocal microscopy. Fixed cells were imaged with Plan-Neofluar 40×/1.30 Oil objective. Time lapse imaging of living cells was done at 2fps with a Plan-Apochromat 63×/1.40 Oil objective and using the line switching mode to avoid any fluorescence channel crosstalk and to minimize acquisicial delay between channels. Images in Supplementary Fig. 6 and Supplementary Movie 2 are 332 × 268 with 85 nm pixel dimension. Channels were combined and movies were annotated with the Fiji distribution of ImageJ.

### Recombinant proteins purifications

YARS1 protein purification for actin co-sedimentation assays: Sequence verified, human full-length YARS1, encoding the wild type or the E196K substitution, was carried on a pCRT7/NT-TOP expression plasmids carrying an Xpress™–6xHis tag, and used to transfect BL21(DE3)pLysS *E. coli* competent cells, grown on LB/carbenicillin plates at 37 °C overnight. Colonies from this plate were scraped to grow a 250 ml liquid preculture (37 °C overnight). The preculture was then added to 1 l LB/ carbenicillin. This 1.25 l culture was grown for ~ 4 h or to an $OD_{600}$ of 0.7–0.8 at 37 °C. The culture was incubated on icy water for ~30 min, before isopropyl β-D-thiogalactopyranoside (IPTG) was added to a final concentration of 0.2 mM to induce the protein expression for 4 h at 37 °C. The cells were pelleted by centrifugation at 5000 *g* for 10 min, resuspended in ddH$_2$O, snap-frozen in liquid nitrogen, and kept at −80 °C. Bacterial pellets were thawed in 1 volume of 2X lysis buffer (100 mM Tris, 600 mM NaCl, 40 mM Imidazole, pH 7.5), supplemented with 5 mM β-mercaptoethanol, aqueous protease inhibitor, HALT protease inhibitor and Pepstatin A, with gentile agitation at room temperature, followed by sonication of 4 cycles (30 s on, 30 s off, 80% power output). CHAPS (0.1% final concentration) was added to the lysate before incubating for 10 min on ice. The soluble supernatant and the insoluble pellet were separated by centrifugation for 10 min at 12,000 *g*. The supernatant was run through PVDF syringe 0.22 μm filters before loading onto a buffer-equilibrated 1 ml HisTrap HP column (Thermo Fisher). Elution was carried out in buffer containing 50 mM Tris, 20 mM NaCl, 400 mM Imidazole, pH 7.5, supplemented with 0.1% CHAPS and 10% glycerol. The fractions containing the proteins were combined, ion exchange buffer (20 mM Tris, 10% glycerol, pH 8.5) was added and sample was loaded onto a Resource Q ion exchange column. The protein-containing eluates (in 20 mM Tris, 1 M NaCl, 10% glycerol, pH 8.5) were concentrated to 500 μl and loaded on a Superose column for a final step of gel filtration in 20 mM Tris, 100 mM NaCl and 10% glycerol, pH 7.6. The Xpress™–6xHis tagged YARS1 enriched fractions were collected and concentrated, the purified protein was aliquoted, snap-frozen and stored at −80 °C.

Purification of recombinant YARS1 proteins for aminoacylation assay: YARS1$^{WT}$ and the YARS1$^{E196K}$ mutation were cloned into pET-20b vector (Novagen). Construct were transformed into BL21(DE3) cells. Cultures were grown overnight to saturation in LB medium containing 100 μg/ml ampicilin. The overnight culture was diluted 1/100 in LB medium and grown at 37 °C. Isopropyl ß-D-thiogalactopyranoside (IPTG) was added to a final concentration of 0.4 mM at OD600 of 0.7, and then the cells were grown at 37 °C for 3 h. The induced cells were collected by centrifugation at 4000 rpm for 20 min. The pellet was resuspended in 5 volumes (v/w) lysis buffer (20 mM Tris/HCl, 300 mM NaCl, 10 mM Imidazole, 1 mM phenyl-methyl-sulphonyl-fluoride, pH 8.0). The cells were disrupted by microfluidizer (M-110P) and the lysate was clarified by centrifugation at 35,000 *g* for 30 min. TyrRS and mutant proteins were purified using Ni-NTA beads (Qiagen) and a HiLoad 16/60 Superdex 200 prep grade column (GE Healthcare). All purification steps were carried out at 4 °C or on the ice.

Recombinant GST-tagged PLS3 cloning and purification: For generation of GST-tagged PLS3, the cDNA of PLS3 was cloned into PGEX4T3 vector using Not1 and Mlu1 restriction sites. For the production of GST-PLS3 recombinant protein, the bacterial cultures were grown in the presence of 50 μg/ml Ampicillin. When the OD reached 1.2, the culture was induced using 1 mM IPTG concentration for 4 h at 25 °C. Next, the bacterial cells were collected and lysed in lysis buffer containing 50 mM Tris-HCl (pH:8.0) 100 mM NaCl, 5 mM Beta Mercaptoethanol (BME), and 1% Triton-100. The GST-PLS3 was purified using GST beads and the fraction of bound protein was eluted using 50 mM concentration of reduced glutathione in elution buffer (50 mM Tris-HCl (pH:7.4) 100 mM NaCl, 5 mM BME and 15% glycerol. The purified GST-PLS3 was studied using SDS page and Coomassie staining.

Recombinant GARS1 purification: Expression and purification of GARS1[WT] was performed as described in ref. [68]. Briefly, C-terminal His-tagged human GARS1[WT] was cloned into pET21b vector (Novagen) and expressed in *Escherichia coli* BL21 (DE3) host cells at 25 °C. The protein was purified by Ni-NTA agarose affinity column followed by ion-exchange monoQ column and size-exclusion column Superdex 200 (GE Healthcare).

Recombinant HARS1 and DARS1 were purchased from ProSpec Tany. Recombinant Human HARS1 (product code ENZ-268) was produced in *E.coli* as a single, non-glycosylated, polypeptide chain having a molecular mass of 55 kDa. Recombinant DARS1 (product code ENZ-591) was also produced in *E. coli* as a single polypeptide chain containing 521 amino acids, having a molecular mass of 59.3 kDa, and it is fused to a 20 amino acid His-tag at N-terminus. Both proteins were purified by proprietary chromatographic techniques.

Rabbit skeletal muscle actin was purified as described[69]. Fluorescent rabbit muscle actin labeled on Cys[374] with OG maleimide (Life Technologies, Carlsbad, CA) was generated as described[70].

## In vitro binding assays

Recombinant GST-tagged PLS3 (2 μg) was co-incubated with recombinant His-YARS1 proteins (2 μg) or IgG beads (GE Healthcare) in PBS with 0.05% Tween-20 at pH 7.4 for 1 h on a rotary shaker at 4 °C. YARS1 was immunoprecipitated with YARS antibody (H00008565-M02, Abnova). The input and IP fractions were analyzed via Western blotting using the YARS1 antibody and rabbit polyclonal PLS3 antibody (ab137585). Following the Dynabeads manufacturer's instructions, recombinant His-YARS1 proteins were preincubated individually with GST-PLS3 or α-actinin (Cytoskeleton) in the presence of prewashed magnetic beads (Dynabeads) for 20 min at room temperature with constant agitation. The binding/washing buffer was 50 mM NaP, pH 8.0, 300 mM NaCl 0,01% Tween-20. Beads were collected via magnet, the supernatant was discarded, and the beads were washed four times. His-YARS1 was eluted with the low pH elution buffer (150 mM Imidazole, 50 mM NaP pH 8.0, 300 mM NaCl, 0.01% Tween-20) and the eluate was mixed with protein sample buffer. The input and pull-down fractions were analyzed with SDS-PAGE. The gels were stained with Coomassie R250 dye-based reagent (Pierce).

## YARS1/PLS3 F-actin co-pelleting assay

Recombinant His-tagged YARS1 proteins at three different concentrations were co-incubated with preassembled F-actin at 2 μM and 1 μm GST-tagged PLS3 for 30 min, then samples were subjected to high-speed pelleting at 90,000 g for 1 h at 24 °C. Supernatants were carefully removed, sample buffer was added to supernatant and pellet, and equal volumes of both fractions were analyzed by SDS-PAGE and Coomassie R250 dye-based reagent (Pierce).

## Actin filament binding- and bundling assays

High-speed pelleting to determine binding to F-actin for the YARS1 recombinant proteins was performed by incubating increasing concentrations of preassembled F-actin with 500 nM YARS1 (30 μl total reaction) for 30 min at room temperature, followed by 15 min pelleting at 90,000 g (4 °C) in a TLA-100 rotor (Beckman). The entire supernatant (~30 μl) was removed to a new tube, to which 2x Laemmli Buffer was added. The pellet was dissolved by vigorous pipetting in 60 μl Laemmli buffer. Half of the supernatant and the pellet (30 μl) were run on an SDS gel, stained with Coomassie solution, and imaged on a ChemiDoc Imaging system (Bio-Rad). Images were analyzed in ImageJ, by subtracting background, and further using the Gel function in the Analyze menu. We plotted the fraction of proteins in the pellet and used GraphPad Prism to fit the data into Michaelis-Menten nonlinear curve. Low-speed pelleting to determine F-actin bundling capacity for the YARS1 proteins was performed five different times by incubating increasing concentrations of YARS1 with preassembled F-actin at 2 μM

concentration in a 30 μl reaction for 30 min at room temperature. The reactions were then pelleted for 15 min at 14,000 g in a tabletop centrifuge precooled to 4 °C. Separation of supernatants and pellets, gel electrophoresis and staining, and image analysis were conducted identically to the high-speed pelleting assay.

## G-actin sequestering assay

To test for putative G-actin sequestering functions of YARS1, 16 μM muscle actin (Cytoskeleton™) in General Actin Buffer was incubated with α-actinin and BSA (both controls at 2 μM final concentration) and 2 μM YARS1 proteins for 30 min at room temperature. All proteins were also tested alone, in the absence of G-actin. After the 30 min incubation, Actin Polymerization Buffer (Cytoskeleton™) was added to each tube with additional incubation at room temperature for 30 more minutes. All tubes were spun at 150,000 g for 1.5 h at 24 °C. Separation of supernatants and pellets, gel electrophoresis, and staining were identical to the ones described in the other pelleting assays.

## TIRF microscopy of F-actin filaments

Coverslips (24 × 60; Fisher Scientific) were cleaned/etched by sonication in the following solutions: 60 min in detergent, 20 minutes in 1 M KOH, 20 min in 1 M HCl, and 60 min in ethanol. Next, the coverslips were extensively washed with ddH₂O and dried in a stream of N₂. A freshly prepared solution of 80% ethanol (pH 2.0), 2 mg/ml methoxy-poly (ethylene glycol)-silane, and 2 μg/ml biotin-poly (ethylene glycol)-silane (Laysan Bio Inc.) was applied and spread on the cleaned coverslip (~175 μl), which was further incubated for 16 h at 70 °C. Flow cells were assembled before use: the coated coverslips were rinsed extensively with ddH₂O, dried in an N₂-stream and attached to a flow chamber (Ibidi) with double-sided tape, and completely sealed with epoxy resin for at least 15 min. Before starting the imaging, the flow cell was incubated 3 min in HEK-BSA (20 mM HEPES pH7.5, 1 mM ethylenediaminetetraacetic acid (EDTA), 50 mM KCl, 1% BSA), then equilibrated with 1xTIRF buffer (10 mM imidazole, 50 mM KCl, 1 mM MgCl₂, 1 mM EGTA, 0.2 mM ATP, 10 mM dithiothreitol (DTT), 15 mM glucose, 20 μg/ml catalase, 100 μg/ml glucose oxidase, and 0.5% methylcellulose (4000 cP, pH 7.5)). F-actin polymerization was initiated by flowing in 2 μM 10% OG-labeled G-actin in TIRF buffer into the flow chamber and monitored until filaments reached at 10–15 μm length. The samples were imaged briefly before washing out free actin monomers and flowing in YARS1 or TIRF buffer alone. We recorded three proximal, nonoverlapping fields-of-view in TIRF mode for a total time of 20 min, at 0.1 Hz frequency. Single color, time-lapse TIRF microscopy was conducted on a Nikon-Ti200 inverted microscope equipped with a 150 mW Argon Laser (Mellot Griot, Carlsbad, CA), a TIRF-objective with NA of 1.49 (Nikon Instruments Inc., New York, NY) and an EMCCD camera (Andor Ixon, Belfast, Northern Ireland), using the optimal focus via the perfect focus system. Pixel size is 0.178 μm.

We analyzed TIRF recordings within 15 min after flow-in of buffer or YARS1 protein. The quantified data presented in Fig. 2h and Supplementary Fig. 5g show two representative experiments (one at 0.5 μM YARS1 and one at 2 μM YARS1) in which all three samples (YARS1[WT], mutant YARS1[E196K], and buffer alone) were imaged from a single coated coverslip with multiple flow chambers. These experiments were repeated several times and in each case F-actin bundle and cable formation was induced by YARS1; however, there was frequent "stickiness" observed in the presence of YARS1 that prevented robust scoring of real-time events with TIRF.

The movies were analyzed using the ImageJ software[60], by applying the following workflow: images underwent background subtraction (rolling ball radius of 10 pixels), subsequent segmentation using Renyi's entropy autothresholding, and masking the regions of interests (ROIs). This segmentation approach captured single filaments, areas of adjacent unbundled filaments, bona fide bundles, and thick cables. Mean gray value was measured in the masked ROIs, which

reflects the change of fluorescence due to "condensation" of the dispersed filaments at time 0 (the flow-in of YARS1) into bundle and cables over time. The fold change of the mean gray value (MGV) in the segmented areas was obtained by dividing the MGV at the different time points with the MGV at time 0 (the flow-in of buffer in CTRL or YARS1). This type of quantification is presented in Fig. 2h and Supplementary Fig. 5g.

## Aminoacylation activity of YARS1 in the presence of F-actin

The aminoacylation assays were performed at room temperature with 50 mM Hepes (pH 7.5), 20 mM KCl, 5 mM MgCl2, 4 mM ATP, 2 mM DTT, 4 μg/mL pyrophosphatase, 20 μM cold L- tyrosine, and 1.34 μM [3H]- tyrosine (1 mCi/mL) as the assay solution. Total yeast tRNA (1 μg/μL) was mixed with the assay solution, and the reaction was initiated by adding YARS1 protein with or without F-actin into the mixture. At varying time intervals, 5-μL aliquots were applied to a MultiScreen 96-well filter plate (0.45-μm pore size hydrophobic, low-protein-binding membrane; Millipore), which is prewetted with quench solution containing 0.5 mg/mL DNA and 100 mM EDTA in 300 mM NaOAc (pH 3.0).

## Western blot analysis

For protein expression level experiments on *Drosophila* material, we used adult fly head extracts in RIPA buffer. Protein concentration was determined using the Biorad Bradford protein assay. The following primary antibodies were used: mouse monoclonal β-actin (1:10,000, Abcam), mouse monoclonal actin (JLA20, 1:50, Developmental Studies Hybridoma Bank), mouse monoclonal PLS2 (1:1000, Abcam ab83496) and rabbit polyclonal PLS3 (1:1000, Abcam ab137585). HRP-labeled secondary antibodies were used (Jackson ImmunoResearch Laboratories Inc.). A digital 16-bit image of the chemiluminescent signal was acquired with ImageQuant™ LAS 4000 (GE Healthcare Life Sciences). The expression levels of *UAS-PLS3* and *UAS-PLS2* were determined upon protein extraction from heads of *nSyb-Gal4 > UAS-PLS3* and *nSyb-Gal4>UAS-PLS2* flies. Images were analyzed in ImageJ[60], by subtracting background, and further using the Gel function in the Analyze menu.

SH-SY5Y cells were lysed in lysis buffer (150 mM NaCl, 1% NP-40, 0.5% sodium deoxycholate, 0.1% SDS, 50 mM Tris, together with protease inhibitor) for 20 min on ice and cleared by centrifugation for 10 min at 14.000 rpm. Protein concentration was determined using the Pierce BCA protein assay kit (ThermoFisher Scientific). Cell lysates were boiled for 10 min at 95 °C in reducing Laemmli buffer (Life Technologies) supplemented with 100 mM 1,4 DTT. Proteins were separated on Bio-Rad 10% Mini-PROTEAN® TGX™ Precast Protein Gels (Bio-Rad) and transferred on nitro-cellulose membrane by semi-dry transfer on the Trans-Blot® Turbo™ Transfer System (Bio-Rad). Blocking of the membrane was performed using 5% milk diluted in PBS, supplemented with 0.1% Tween 20. Afterwards, the following primary antibodies were used: mouse monoclonal YARS1 (1:2000, H00008565-M02, Abnova), rabbit monoclonal Flag antibody (1:500, F2555, Sigma-Aldrich), and anti-tubulin (1:10,000, ab7291, Abcam).

Horseradish peroxidase-labeled goat anti-mouse IgG1 (1:10,000, 1070-05, sanbio B.V.) was used for chemiluminescence detection. Exposure of the membrane was performed with ECL (11527271, Fisherscientific) in AI600 chemiluminescent imager (GE Healthcare). Images were further analyzed and quantified with ImageJ[60]. One-Way Anova statistical analysis was used to compare protein expression levels.

## Reporting summary

Further information on research design is available in the Nature Portfolio Reporting Summary linked to this article.

## Data availability

All data generated and/or analyzed during this study are included in this article (and its supplementary files). The source data underlying Figs. 2–6, and Supplementary Figs. 1, 2, 4, 5, 7–10 are provided as a Source Data file. The mass spectrometry data are deposited at the PRIDE database under accession number PXD037630. All other relevant data are available from the corresponding author on reasonable request. The human biomaterials are available subject to MTA. Databases and softwares relevant to the study include ImageJ [https://imagej.nih.gov/ij/]; FlyBase [http://flybase.org/]; PANTHER [http://pantherdb.org/]; the quantitative proteomics software package MaxQuant [https://www.maxquant.org/]; ClustalOmega [https://www.ebi.ac.uk/Tools/msa/clustalo/]; DroID [http://droidb.org/]. Source data are provided with this paper.

## Code availability

Relevant code and classifiers (e.g., ImageJ macro scipts) are available from the corresponding author on reasonable request.

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

## Acknowledgements

We thank L. Svensson, I. Pintelon, J. P. Timmermans for use of the SEM; S. Munck for advice on SIM; J. Gettmans for PLS3 and PLS2 constructs; G. Hoeprich and C. Fees for providing actin reagents and assistance with F-actin bundling assays; N. Wei for providing recombinant GARS1^WT protein; E. Ydens for advice on expression levels determination; T. Ooms for the excellent technical assistance; members of the Verstreken and Godenschwege lab for help and advice on the NMJ and GFS electrophysiology, respectively; P. De Rijk for calculating the YARS1 expression levels in the YARS1^CMT fly models; Pieter Van de Walle for the help with the artistic drawing; V. Mitev for the productive discussions. This work was supported in part by funding from the University of Antwerp to A.J. and V.T. The Fund for Scientific Research-Flanders: G048220N and G0A2122N to A.J., G041416N to V.T. The Belgian Association Against Neuromuscular Diseases grants to A.J., M.L.P.E., V.T. The American Muscular Dystrophy Association grant 175816 to A.J. The French Muscular Dystrophy Association grants 16179 and 23708 to A.J., and Trampoline grant 21869 to M.L.P.E. The National Institute of Child Health and Human Development R01HD050725 to T.A.G. The National Institute of Neurological Disorders and Stroke: 1R21NS120123-01A1 to X.L.Y. and A.J., and NS116375 to A.A.R. The National Institute of General Medical Sciences R35 GM134895 to B.L.G. The German Research Foundation [Wi945/17-1, CRC1451 (project-ID 431549029 – A01), GRK1960 (project ID 233886668) and FOR 2722 (project ID 407176282) to B.W. The Bulgarian National Science Fund and the Bulgarian National Plan for Recovery and Resilience (grant BG-RRP-2.004-0004-C01 to A.J. and I.T). The European Research Council (ERC) under the European Union's Horizon 2020 research and innovation program under the Marie Skłodowska-Curie grant agreement 956185 (SMABEYOND) to B.W., and the ERC Consolidator grant to P.V. Center for Molecular Medicine Cologne (project C18) to B.W. Fund for Scientific Research-Flanders Ph.D. fellowships to B.E., R.L.G., D.A., L.M. Fund for Scientific Research-Flanders postdoctoral fellowship to M.L.P.E. Boehringer Ingelheim Funds travel grant to B.E.

## Author contributions

Conceptualization: B.E., B.A., R.L.G., M.L.P.E., P.C., A.J. Methodology: B.E.,. B.A., L.M., M.L.P.E., S.H., R.L.G., L.A.S., S.B., P.V., A.K., I.T., X.L.Y., B.W., A.A.R., V.T., B.L.G., T.A.G., A.J. Investigation: B.E., B.A., L.M., M.L.P.E., S.H., R.L.G., L.A.S., L.S., L.L., D.A., A.K. Visualization: B.E., B.A., L.M., M.L.P.E., S.H., R.L.G., L.A.S., L.S., L.L. Funding acquisition: A.J., V.T., M.L.P.E., B.W., B.L.G., A.A.R. Project administration: A.J. Supervision: X.L.Y., B.W., A.A.R., V.T., B.L.G., T.A.G., P.V., A.J. Writing – original draft: B.E., A.J. Writing—review & editing: all authors contributed to the reviewing and editing of the final version of the manuscript.

## Competing interests

The authors declare no competing interests.

## Additional information

[1]Center for Molecular Neurology, VIB, University of Antwerp, 2610 Antwerpen, Belgium. [2]Department of Biomedical Sciences, University of Antwerp, 2610 Antwerpen, Belgium. [3]Department of Biology, Brandeis University, Waltham, MA 02453, USA. [4]Neuromics Support Facility, VIB Center for Molecular Neurology, VIB, 2610 Antwerp, Belgium. [5]Neuromics Support Facility, Department of Biomedical Sciences, University of Antwerp, 2610 Antwerp, Belgium. [6]Institute of Human Genetics; Center for Molecular Medicine Cologne; Center for Rare Diseases Cologne, University Hospital of Cologne; University of Cologne, 50931 Cologne, Germany. [7]Department of Molecular Medicine, The Scripps Research Institute, La Jolla, CA 92037, USA. [8]Department of Biological Sciences, Florida Atlantic University, Jupiter, FL 33458, USA. [9]Department of Neurology, Medical University-Sofia, 1431 Sofia, Bulgaria. [10]Department of Cognitive Science and Psychology, New Bulgarian University, 1618 Sofia, Bulgaria. [11]Department of Human Genetics, KU Leuven, 3000

Leuven, Belgium. [12]VIB-KU Leuven Center for Brain & Disease Research, 3000 Leuven, Belgium. [13]KU Leuven, Department of Neurosciences, Leuven Brain Institute, Mission Lucidity, 3000 Leuven, Belgium. [14]Department of Medical Chemistry and Biochemistry, Medical University-Sofia, 1431 Sofia, Bulgaria. [15]Present address: Division of Endocrinology and Metabolism and Department of Neuroscience, University of Virginia, Charlottesville, VA, USA. [16]Present address: Frontiers Media SA, Lausanne, Switzerland. [17]Present address: Helsinki Institute of Life Science, Institute of Biotechnology & Faculty of Biological and Environmental Sciences, University of Helsinki, Helsinki, Finland. [18]Present address: Department of Neurobiology, University of Utah, Salt Lake City, UT, USA. [19]Present address: School of Public Health (Shenzhen), Sun Yat-Sen University, Guangdong, China. [20]Present address: Center for Social and Clinical Research, National Minority Quality Forum, Washington, DC, USA. [21]Present address: Max Planck Institute of Immunobiology and Epigenetics, Freiburg, Germany. ✉e-mail: Albena.Jordanova@uantwerpen.vib.be

