## [Peer Review File · Nature Communications]

Tyrosyl-tRNA synthetase has a non-canonical function in actin bundlingREVIEWER COMMENTS

Reviewer #1 (Remarks to the Author):

The paper by Ermanoska et al. reports on a novel actin-bundling function of the tyrosyl-tRNA synthetase (YARS1), which was detected in an unbiased genetic screen in *Drosophila*. The authors show that genetic variants which cause Charcot-Marie-Tooth inherited neuropathy (CMT) result in dysfunction of the actin cytoskeleton organization in *Drosophila* neurons and patient-derived skin fibroblasts. Genetic modulation of F-actin organization resulted in improved electrophysiological and morphological features in *Drosophila* neurons, which were expressing CMT-causing YARS1 mutations. Similar results were detected in flies expressing a neuropathy-causing glycyl-tRNA synthetase (GARS1) variant. The authors conclude that YARS1 is an evolutionary-conserved F-actin organizer and that the actin-cytoskeleton axis may be also implicated in other tRNA-synthetase-related CMT.

1. I am a bit puzzled by the large variety of molecular functions which have been reported recently to be disturbed by autosomal dominant variants in CMT-causing tRNA synthetase mutations. For example, recent studies from the same group reported in *Nature Communications* that CMT-causing YARS1 variants induce unique conformational changes, leading to aberrant interactions with transcriptional regulators in the nucleus, leading to transcription factor E2F1 hyperactivation. Pharmacological inhibition of YARS1 nuclear entry prevented phenotypes of CMT in the *Drosophila* model. How can the authors link these previous data to the current results?
2. This paper also state, that the actin cytoskeleton has a role in GARS1-related CMT, where also recent excellent papers in *Science* detected that there is a very important role of the integrated stress response, which may be triggered by low level of charged tRNA. For GARS1 there is also very strong data published in *Nature* to support the role of abnormal binding to neuropillin 1.
3. As several tRNA synthetases are linked to CMT, it is likely that there is a common main mechanism behind this phenotype. It would be good to have a broader view on how this novel actin organising role can be fitted to the puzzle of the molecular mechanism of CMT-causing tRNA synthetases. Are the different mutations act on different functions? Or even single variants can impair more of the suggested molecular mechanisms? Clarifying these points would be very useful if we think about developing treatments for patients with CMT.
4. I suggest to add a paragraph about translational aspects.
5. It would be useful to provide a summary figure on the different molecular functions, different binding partners which have been implicated in these defects. Also, the authors should clarify whether the very same *Drosophila* models and fibroblasts were used in the other studies, where the relevance of another pathway was highlighted.
6. I suggest to use the current name of the genes YARS1, GARS1 through the manuscript.

Reviewer #2 (Remarks to the Author):

Ermanoska et al investigate interactors of Tyrosyl-tRNA synthetase (YARS) that could contribute to the phenotypical severity of mutations found causative for Charcot-Marie-Tooth peripheral neuropathy. To this end, they use a *Drosophila* fly model in which defects in the organization of the eye are used as a read-out to screen for genetic interactors in the background of the E196K YARS mutation. They find that overexpression of Fimbrin, an actin bundler, enhances disorganization of the eye in the context of E196K expression but not on its own or in conjunction with YARS wt. This suggests actin organization as a potential site of lesion in CMT, specifically in the context of YARS mutants. They confirmed physical interaction between YARS and five actin binding proteins and visualize YARS-induced actin bundling *in vitro*. In patient fibroblasts, F-actin patterns were irregular and defective in cytoskeleton organization. In the context of neuromuscular junctions that are primarily affected in CMT, YARS affected acting distribution and synaptic vesicle mobility. The phenotypic changes in neuronal actin are also found upon GARS CMT mutations, suggesting that this is a shared mechanism in CMT. Reduction of Fimbrin restored giant fiber physiology, suggesting that targeting Fimbrin might be a potential strategy to alleviate CMT symptoms.

The manuscript is concise to the point where crucial information can be easily missed. Background information on actin and actin bundling, especially in the context of neurological disorders, would help put findings into perspective – the introduction does not mention actin and not much information on actin and actin bundling outside of the interaction with YARS is provided. Therefore, it is difficult to assess the impact of YARS compared to other actin organizers and in consequence the impact of the findings reported here. I appreciate that actin is well-investigated and thoroughly reviewed elsewhere but it would help making the manuscript more approachable for a general audience. The findings seem exciting, impactful, and relevant but the presentation hinders accessibility. While new experiments might not be necessary, I strongly recommend additional schemes and a more expansive introduction and discussion.

Introduction: There is one sentence on regarding actin modulators in neuropathies towards the end of the discussion – this could be elaborated on and moved to the introduction. What is the function of actin bundling and actin bundles vs growing actin filaments under physiological conditions and how are these affected in other neurological disorders? Additional information on actin bundling would be appreciated: how are bundles defined, what are properties of known bundling proteins, is this ATP dependent, what is the usual ratio between actin and actin bundling proteins... It might seem trivial, but the general audience (including this reviewer) would appreciate it if background information on actin would be provided.

Figure 1: What impacts fly eye organization and how is it a good platform to study DI-CMT, which predominantly affects motor neurons? Citation 13 mentions an eye phenotype in one of the supplementary figures but focuses more on climbing and jumping abilities, which are a more natural choice for a disease that affects motor neurons. Are any of the 2.15% of EPs that gave retinal phenotypes actin-associated genes? What is the degree of YARS expression over endogenous background?

Figure 2: It would help to have a schematic on the actin bundling process and the difference between binding and bundling – is the difference between the assays only in the centrifugation speed? Is the concentration at which YARS induces bundling high or low compared to other bundling proteins and do the observed concentration fall within the expected physiological range of aaRS? Figure 2a and 2c should be separated and displayed next to their quantification – the current arrangement is rather confusing.

In light of the later findings, how does the identification of Coro, dPod, and IKKe as additional genetic modifiers fit? Are these expected to have overlapping functions with YARS or Fimbrin?

S4: Was PSL3 the only identified Plastin due to its preferential expression by HEK293 cells? That should be easy to check using expression databases of HEK cells. Was actin itself found in the YARS interactome? The findings in Figure 2 suggest that it directly interacts with YARS?

Figure 3: Figure 3 is clear, with informative schematics and intuitive data presentation. It would be helpful if this level of clarity could be achieved for Figure 1 and 2. It is touched upon in the discussion but how do systemic changes in actin lead to the tissue-specific phenotype observed in CMT patients?

Figure 4b: The figure legend should at least briefly describe the criteria used for quantification of Lifeact distribution or consider moving S6A to the main figure. In the provided images, the redistribution is more striking towards the inside of the bouton than close to the rim – how was the distribution at 700 nm chosen as a criteria?

Figure 5: How can a similar rescue by Fimbrin in GARS mutants be explained if the suggested mode of action is through interaction between YARS and actin?

Mass spectrometry data should be deposited and made accessible after release (for example through PRIDE).

Page 6, missing word: We demonstrated that under stress conditions A fraction of ... (a missing)

Reviewer #3 (Remarks to the Author):

In this paper, the authors identified *Drosophila* Fim (Fimbrin/CG8649) in an unbiased screen for enhancers of dominant mutations in the tyrosyl-tRNA synthetase (YARS). This and other tRNA-ligases are mutated in Charcot-Marie-Tooth neuropathies (CMTs). Functionally, YARS exhibited an unexpected actin-binding and bundling activity, which suggests that misregulation of the actin cytoskeleton in mutant YARS proteins could be a driving force in neurodegeneration. While this hypothesis is compelling the data provided do not fully support this idea. I have therefore a number of questions and suggestions to improve the manuscript.

Major comments:

1) The authors screened 557 lines containing enhancer-promotor P-element insertions (EPs) and reference Rorth PNAS 1996 as a source (page 2). However, this paper mentions only 352 target lines (p. 12419). The M+M section reveals that the screened lines are derived from 4 different sources (page 14). This prevents precise tracking of the lines actually screened. The authors should therefore provide a complete list of the lines used for reproducibility purposes.

2) The authors state (p. 3) that they ultimately identified a single line BDSC_14274, which they call FimEP (p. 3) but sometimes also EPFim (Suppl. FigS1a). However, the Bloomington stock center lists CG5445 and not Fim as the parent gene under the accession number BDSC_14274. This should be clarified in the text, e.g. that the element is oriented to drive Fim but not CG5445. According to FlyBase, at the same location, there is also a second EP line, P(EP)FimG10929. Why was this line not detected in the screen? The authors should test if it behaves similar to BDSC_14274.

3) Various effects of mutant YARS on actin organization is shown in different experiments. Fig. 2 shows that mutant YARS binds and bundles F-actin faster than wild-type YARS in biochemical assays. Fig. 3 shows perturbed F-actin cables and disrupted stress fibres in patient-derived fibroblasts in culture. Fig. 4 shows altered distribution of Lifeact::Ruby at *Drosophila* larval neuromuscular junctions (NMJs), and Fig. 5 shows YARS-induced constrictions in the giant fibre system. All these findings are therefore individual observations in different systems but it is unclear if they are causally connected. For example, are the constrictions in the giant fibre system caused by defects in the organization of F-actin, or are they caused by the observed defects in synaptic vesicle? Also, F-actin re-organization should lead to constrictions at NMJs, too, but synaptic boutons appear unchanged in Fig. 4a. In the end, it should be possible to generate CMT-like phenotypes, which are characterized by progressive distal muscle weakness.

4) For the interpretation of the detected genetic interaction between YARS and Fimbrin in the eye, it would be important to demonstrate that YARS and Fim also functionally co-operate in peripheral nerves and synergistically generate CMT-like phenotypes. While the authors show that endogenous YARS localizes to synaptic boutons, the distribution of Fim is less clear, although several exon-trap lines seem to exist. Is endogenous Fim co-expressed with YARS in motoneurons or at NMJs? Would co-expression of Fim and YARS increase actin-bundling activity (biochemically, S2 cells), completely dissolve Lifeact::Ruby at NMJs or augment constrictions in the giant fibre system?

5) While the F-actin marker Lifeact::Ruby stains the "bouton border proximity" (Fig. 4), UAS-GFP-actin distributes in a defined punctate pattern at presynapses of larval NMJs (Pawson et al., JNS 28, 11111 (2008), there Fig. 3). Is there any explanation for this differential localization? Would YARS proteins also be able to re-localize GFP-actin and dissolve these puncta?

6) The reason for seeding normal and patient-derived fibroblasts on Y-shaped microstructures rather than on normal tissue culture dishes is not explained (Fig. 3, p. 4, p.17). Why should fibroblasts be informative if disease occurs in neurons and axons? Even if patient-derived neurons are not available, it would be possible to transfect neurons with wild-type and mutant constructs

and determine the consequences on F-actin organization. In addition, Fig. 3a and 3e show patient-derived fibroblasts but speckle-like accumulations, a characteristic phenotype of mutant YARS, is visible only in Fig. 3a. Reduced co-localization with F-actin stress fibres in YARS-mutant fibroblasts is not really visible in the images shown in Fig. 3e and might require higher magnifications (40x-100x objectives).

Minor points:

a) Fig. 1: Insets are too small and should be enlarged, as it is not possible to judge bristle arrangement in printouts. In addition, Fig. 2f appears too dark.

b) Legend to Fig. 5 (page 14): panel "f" should be "e"

c) There are different classes of CMT. The authors should specify to which class they are referring in the paper (neuronal or glial forms).

d) Suppl. Fig. S2b: The meaning of the dashed line should be indicated in the legend.

Suppl. Fig. S6: The control image in S6b is identical to the control image in Fig. 4a and should be replaced.

Suppl. Fig. S8 seems unnecessary.

Suppl. Fig. S9 FasII-staining in S9c does not represent the average distribution of FasII at NMJs and should be replaced.

RESPONSE TO REVIEWERS COMMENTS

We would like to thank all three Reviewers for their most helpful input and comments. We believe the paper has been strengthened as a result of addressing them. Experimentally, we have performed a new series of *in cellulo* analyses in human SH-SY5Y neuroblastoma cells further demonstrating the disorganization of actin cytoskeleton upon YARS1^{CMT} expression. We also expanded our *in vivo* evidence about actin rearrangements in the nervous system of the *Drosophila* YARS1^{CMT} model, which align with the effects of the actin bundler PLS3. Moreover, we provided biochemical evidence for direct binding between F-actin and three additional aminoacyl-tRNA synthetases (AARSs), thus extending the possibility of actin involvement in AARSs-associated axonopathies. As a result, we included two additional co-authors who contributed to the revision experimentally. To improve the comprehension of the text, we enriched the Introduction and Discussion of the manuscript and provided more context on the described results. In the revised manuscript, all major textual changes are highlighted in red, new or updated figures are framed with blue boxes.

Responses to the comments of Reviewer #1 (Remarks to the Author):

The paper by Ermanoska et al. reports on a novel actin-bundling function of the tyrosyl-tRNA synthetase (YARS1), which was detected in an unbiased genetic screen in *Drosophila*. The authors show that genetic variants which cause Charcot-Marie-Tooth inherited neuropathy (CMT) result in dysfunction of the actin cytoskeleton organization in *Drosophila* neurons and patient-derived skin fibroblasts. Genetic modulation of F-actin organization resulted in improved electrophysiological and morphological features in *Drosophila* neurons, which were expressing CMT-causing YARS1 mutations. Similar results were detected in flies expressing a neuropathy-causing glycyl-tRNA synthetase (GARS1) variant. The authors conclude that YARS1 is an evolutionary-conserved F-actin organizer and that the actin-cytoskeleton axis may be also implicated in other tRNA-synthetase-related CMT.

1. I am a bit puzzled by the large variety of molecular functions which have been reported recently to be disturbed by autosomal dominant variants in CMT-causing tRNA synthetase mutations. For example, recent studies from the same group reported in Nature Communications that CMT-causing YARS1 variants induce unique conformational changes, leading to aberrant interactions with transcriptional regulators in the nucleus, leading to transcription factor E2F1 hyperactivation. Pharmacological inhibition of YARS1 nuclear entry prevented phenotypes of CMT in the *Drosophila* model. How can the authors link these previous data to the current results?

We acknowledge the Reviewer's remark on the complexity and diversity of functions associated by us and others to the synthetases and their connection to CMT. Our view on this question was outlined briefly in the 3rd paragraph of the original Discussion and we elaborated on this in the 4th paragraph of the revised version.

As stated in this study and in the cited article, we think that the YARS1-associated (and potentially AARS-associated) neurodegeneration is pleiotropic in nature and is due to the interplay of multiple molecular pathways. Our current study adds to the complex picture the contribution of the actin cytoskeleton via the novel actin bundling function of YARS1^{WT} that is enhanced by the CMT mutations. YARS1 translocates to the nucleus in a tRNA- and acetylation-dependent manner upon

oxidative stress^{1,2}, which opens up a possibility that it can modulate the nuclear cytoskeleton, in addition to its role on transcription. It is established that nuclear actin regulates a variety of processes including transcription and transcription regulation, RNA processing and export, dynamic chromatin organization and remodeling, DNA repair, and nuclear envelope assembly³. While a role for YARS1 in nuclear actin remains to be proven in future studies, we envisage at least two plausible scenarios: it can potentially directly modify the organization of the nuclear microfilaments, or it can do this via interactions with other actin binding proteins. Notably, in a previous unbiased retinal degeneration modifier screen we identified two orphan mutant-specific modifiers (*corolla* and *CG15599*) shared between YARS1^{CMT} and GARS1^{CMT} *Drosophila* models⁴. *Corolla* is a nuclear protein involved in the organization of the synaptonemal complex during meiotic division, predicted to interact with actin cytoskeleton and to regulate gene expression. Feature-based function predictions for *CG15599* indicated that it is also a nuclear protein involved in the regulation of transcription and actin binding⁴. Hopefully, with the development of more sensitive molecular tools for visualization of nuclear actin that preserve its versatile organization, functions, and binding partners, we will be able to test these hypotheses in the context of CMT.

2. This paper also state, that the actin cytoskeleton has a role in GARS1-related CMT, where also recent excellent papers in Science detected that there is a very important role of the integrated stress response, which may be triggered by low level of charged tRNA.

In the Discussion, we added additional explanations in the paragraph dealing with the link between actin cytoskeleton, ISR and protein translation, and also in the discussion of the commonalities between the CMT-related AARS.

We believe that our finding that GARS1 (as well as HARS1 and DARS1, Supplementary Fig. 5 i, j) binds to actin could complement the reports that GARS1^{CMT} also triggers ribosome stalling and IRS due to low levels of charged tRNA^{Gly}. Changes in the cellular levels of several different amino acid-charged tRNAs have been reported to induce ISR⁶. At the same time, recent evidence demonstrates that the ISR can be triggered by sensing changes in the F-actin polymerization status too⁷. Thus, the initiation of the ISR observed in the mouse and fly models of GARS1^{CMT} (and YARS1^{CMT}) might be triggered not only via the flux of charged/uncharged tRNAs in the cell (no such data exists for YARS1^{CMT}), but also by a synthetase-actin interaction, which we describe in detail in this manuscript for YARS1, and partially for GARS1 and two more synthetases. Please note also that in the study of Zuko et al., 2021⁸ overexpression of tRNA^{Gly} alleviated - but not rescued- the CMT-specific phenotypes in the GARS1^{CMT} fly model, suggesting the existence of additional molecular players contributing to the neurodegenerative phenotype in the flies. An additional player could be actin and its dynamic composition, as we show that modulating the levels of Fimbrin improves functioning of the Giant Fiber neuronal circuit in the GARS1^{CMT} *Drosophila* model (Figure 6 d, e). Perhaps, the concerted action of both tRNA^{Gly} and Fimbrin would lead to a full rescue effect. Unfortunately, we are limited in proving this hypothesis experimentally, as flies overexpressing tRNA^{Gly} are not publicly available.

For GARS1 there is also very strong data published in Nature to support the role of abnormal binding to neuropillin 1.

We inserted this information in the Introduction of the revised manuscript to prelude the complex mode of action of the synthetases in health and disease. Indeed, Neuropilin 1 was found to bind GARS1^{CMT}, and also Neuropilin 1 was recently identified as an interactor of the CMT-causing alanyl-tRNA synthetase^{9,10}. Notably, by binding integrins, Neuropilin 1 has downstream effect on actin cytoskeleton, which was nicely demonstrated in cell migration assays with tumor cells¹¹. Thus, our findings do not contradict the results presented in the Nature paper of the group of Prof. Xianglei Yang (a co-author of the current manuscript).

3. As several tRNA synthetases are linked to CMT, it is likely that there is a common main mechanism behind this phenotype. It would be good to have a broader view on how this novel actin organising role can be fitted to the puzzle of the molecular mechanism of CMT-causing tRNA synthetases. Are the different mutations act on different functions? Or even single variants can impair more of the suggested molecular mechanisms? Clarifying these points would be very useful if we think about developing treatments for patients with CMT.

We added our view on this point in the second to the last paragraph of the Discussion chapter.

A disease mechanism shared by the synthetases linked to CMT has been hypothesized repeatedly. The discovery of such neurotoxic signaling pathway(s) is highly desirable as its identification might facilitate the development of drugs for a greater number of individuals afflicted with very similar symptoms. This is still an open question in our scientific community. Studies from the Yang lab demonstrate a common conformational opening in four CMT-causing mutant synthetases, but as we stated in the Introduction, the full range of consequences of these defects to the synthetase biology is yet to be determined. Our current study does not exclude previously published findings (Bervoets et al., 2019¹², Zuko et al, 2021⁸, Spaulding et al., 2021¹³ and others), but rather points out that a complex interdependence of the concentration of charged tRNA, actin organization, ISR and the subcellular localization of synthetases might be at the basis of the AARS-induced CMT. This interdependence should be tested experimentally in the future, preferably in a systematic comparative study using unified disease models.

4. I suggest to add a paragraph about translational aspects.

We have linked the translational aspects with the discussion on the potential mechanistic commonalities in the AARS-induced CMT in the second to last paragraph of the Discussion.

5. It would be useful to provide a summary figure on the different molecular functions, different binding partners which have been implicated in these defects. Also, the authors should clarify whether the very same *Drosophila* models and fibroblasts were used in the other studies, where the relevance of another pathway was highlighted.

We appreciate this suggestion. However, we think that a summary figure on the different molecular aspects of AARS^{CMT} known to date might be more suitable for a review article. Therefore, in the Introduction, we direct the readers to two recent reviews published by the Jordanova and Yang labs that describe the current state of art in the field (references 2 and 3).

The *Drosophila* models employed in the current study is the same as in Storkebaum et al., 2009¹⁴ (describing the first YARS1^{CMT} *Drosophila* model), Ermanoska et al., 2014⁴ (establishing common neurodegenerative features and shared genetic modifiers between YARS1^{CMT} and GARS1^{CMT} fly models) and Bervoets et al., 2019¹² (establishing a link between the nuclear function and localization of YARS1 and CMT). This clearly indicates that we are dealing with versatile proteins possessing different properties and excludes the possibility that our findings are a result of the usage of different models. We are using the fibroblast cultures as a model system for the first time in the current study. We included this additional information in the section about fly genetics in the Methods chapter.

6. I suggest to use the current name of the genes YARS1, GARS1 through the manuscript.

Based on the suggestion of the Reviewer we use the gene names (YARS1, GARS1, ...) systematically in the text and figures.

Responses to the comments of Reviewer #2 (Remarks to the Author):

Ermanoska et al investigate interactors of Tyrosyl-tRNA synthetase (YARS1) that could contribute to the phenotypical severity of mutations found causative for Charcot-Marie-Tooth peripheral neuropathy. To this end, they use a *Drosophila* fly model in which defects in the organization of the eye are used as a read-out to screen for genetic interactors in the background of the E196K YARS1 mutation. They find that overexpression of Fimbrin, an actin bundler, enhances disorganization of the eye in the context of E196K expression but not on its own or in conjunction with YARS1 wt. This suggests actin organization as a potential site of lesion in CMT, specifically in the context of YARS1 mutants. They confirmed physical interaction between YARS1 and five actin binding proteins and visualize YARS1-induced actin bundling in vitro. In patient fibroblasts, F-actin patterns were irregular and defective in cytoskeleton organization. In the context of neuromuscular junctions that are primarily affected in CMT, YARS1 affected acting distribution and synaptic vesicle mobility. The phenotypic changes in neuronal actin are also found upon GARS CMT mutations, suggesting that this is a shared mechanism in CMT. Reduction of Fimbrin restored giant fiber physiology, suggesting that targeting Fimbrin might be a potential strategy to alleviate CMT symptoms.

The manuscript is concise to the point where crucial information can be easily missed. Background information on actin and actin bundling, especially in the context of neurological disorders, would help put findings into perspective – the introduction does not mention actin and not much information on actin and actin bundling outside of the interaction with YARS1 is provided. Therefore, it is difficult to assess the impact of YARS1 compared to other actin organizers and in consequence the impact of the findings reported here. I appreciate that actin is well-investigated and thoroughly reviewed elsewhere but it would help making the manuscript more approachable for a general audience. The findings seem exciting, impactful, and relevant but the presentation hinders accessibility. While new experiments might not be necessary, I strongly recommend additional schemes and a more expansive introduction and discussion

We highly appreciate the comment of the Reviewer to expand our main text and provide more background information. We have revised the manuscript accordingly and added a new paragraph

on actin cytoskeleton in the Introduction (the last paragraph), improved the schemes depicting the rationale of our experiments in the different figures, and extended the Discussion chapter (first paragraph) with more comparison between YARS1 and other known actin bundlers.

Introduction: There is one sentence on regarding actin modulators in neuropathies towards the end of the discussion – this could be elaborated on and moved to the introduction.

We moved this sentence to the Introduction and expanded the text to make an entire new paragraph about actin cytoskeleton and its involvement in neuronopathies.

What is the function of actin bundling and actin bundles vs growing actin filaments under physiological conditions and how are these affected in other neurological disorders? Additional information on actin bundling would be appreciated: how are bundles defined, what are properties of known bundling proteins, is this ATP dependent, what is the usual ratio between actin and actin bundling proteins... It might seem trivial, but the general audience (including this reviewer) would appreciate it if background information on actin would be provided.

We appreciate this request of the Reviewer and added dedicated paragraphs in the Introduction (last paragraph) and Discussion (first paragraph) of our manuscript that address (most of) the questions listed.

Figure 1: What impacts fly eye organization and how is it a good platform to study DI-CMT, which predominantly affects motor neurons?

More background information on the retinal degeneration screen is provided in the first paragraph of the Results and we refer to important literature that supports the usage of this experimental paradigm.

Most neurodegeneration-related modifying genes identified in *Drosophila* were recovered from eye screens¹⁵. Importantly, a systematic comparison between the neurodegenerative pathways eliciting toxicity in the fly's developing eye or postmitotic neurons revealed a high overlap and suggested that there are many common pathways of toxicity that govern neurodegenerative cell death of the eye and the brain of *Drosophila*¹⁵. Additionally, our previous work demonstrated that eye-based screens can detect modifiers relevant to CMT (and YARS1^{CMT} in particular)¹².

The adult *Drosophila* compound eye that we used as a screening platform is composed of ~800 independent units, i.e. ommatidia. Each ommatidium has eight photoreceptor neurons that position a rhabdomere near the center of the ommatidium. Defects that affect a single ommatidium will disrupt the positioning of the neighbors. Also, disruption of genes that direct the development of even a single cell within an ommatidium will affect all ommatidia. Of note, rhabdomeres are made of closely packed parallel microvilli that contain high levels of bundled F-actin and associated actin bundling proteins. Thus, this *in vivo* readout facilitates the identification of modifiers relevant to actin cytoskeleton organization. Finally, unlike most organs in the fly, the eye is tolerant of genetic disruption of basic biological processes and is dispensable for survival. In this way, the *Drosophila* eye offers a simple, robust and rapid test for phenotypic modifications.

Citation 13 mentions an eye phenotype in one of the supplementary figures but focuses more on climbing and jumping abilities, which are a more natural choice for a disease that affects motor neurons.

In citation 13 (current citation 18) we indeed describe the dosage-sensitive eye phenotype induced by retinal overexpression of YARS1^{E196K}. In citation 14 (current citation 28) we are building on this result and describe how we employed this phenotype in a large-scale genetic screen for mutant-specific YARS1 modifiers and as a platform to demonstrate that both YARS1^{CMT} and GARS1^{CMT} interact with these modifiers (hence share common neurodegenerative pathways). We employed the retinal screen as an entry point only and later on validated the relevance of the genetic interaction in neuronal paradigms, like the giant fiber neuronal circuit and neuromuscular junctions employed in the current study.

Are any of the 2.15% of EPs that gave retinal phenotypes actin-associated genes?

To the best of our knowledge, none of the EPs that give retinal phenotypes on their own are inserted in genes encoding for actin-associated proteins. With the revised manuscript, we also provide Supplementary Data 1 with a complete list of EP lines screened, along with their full genotype and putative EP-targeted gene(s).

What is the degree of YARS1 expression over endogenous background?

We addressed this question at the mRNA level. The transcript levels of human YARS1^{WT} are on average 17-fold higher, and the human YARS1^{E196K} are 20-fold higher, compared to the endogenous dYARS1 (see graph and table below). There is no statistically significant difference in the expression levels of both human transgenes, indicating that we analysed flies with comparable transgene expression levels and the differences in their phenotype are due to the CMT-causing YARS1 mutation.

To answer this question, we used the deep RNAseq data generated in our previous study establishing the nuclear role of YARS1¹². As stated earlier, the same fly models were used for this and the current study. The sequencing data are publicly available in the GEO database (accession number GSE125311). In short, we isolated total RNA from brains of adult flies with the following genotypes *nSyb-Gal4>+* (*control*), *nSyb-Gal4>2xYARS1-WT* (*WT*), *nSyb-Gal4>1xYARS1-E196K* (*E196K*), that were aged 10 days after eclosion, the point at which mutant flies show locomotor impairment. Relative expression levels of human vs *Drosophila* YARS1 were obtained by aligning all reads to a reference set containing both human and *Drosophila* YARS1 transcript sequence and taking the average depth (using samtools depth) in the region 200 bp. to 1600 bp. of the transcript sequences.

Genotype-sample #	Human transcript_average depth	Drosophila transcript_average depth	ratio
E196K-20	5499,24	234,78	23,42
E196K-21	4168,64	203,64	20,47
E196K-22	4273,37	282,82	15,11
E196K-23	4079,7	207,98	19,62
WT-13	3972,21	224,32	17,71
WT-14	3732,76	278,7	13,39
WT-15	4338,87	202,73	21,4
Control-08	0,09	111,91	0
Control-09	0,41	181,65	0
Control-10	0,09	192,76	0
Control-11	0,72	376,21	0

Figure 2: It would help to have a schematic on the actin bundling process and the difference between binding and bundling – is the difference between the assays only in the centrifugation speed?

We clarified this point in the Results (the section about *in vitro* binding and bundling), and also provided new schemes in Figure 2 a, c, f, and Suppl. Figure 5 b, as requested by the Reviewer.

The speed of centrifugation is one of the key differences between the two types of pelleting assays we used in this study. The high-speed centrifugation pellets all filamentous actin (F-actin) regardless of whether these are single filaments or organized into higher-order structures by bundlers/crosslinkers. The low-speed centrifugation pellets only larger, higher-order F-actin assemblies, including bundles. The other difference between the two assays is that the affinity of YARS1 for F-actin in the high-speed pelleting assays was determined by varying the concentration of F-actin while holding the concentration of YARS1 constant. Reciprocally, in the low-speed pelleting assays the concentration of YARS1 was varied while the concentration of F-actin was held constant.

Is the concentration at which YARS1 induces bundling high or low compared to other bundling proteins and do the observed concentration fall within the expected physiological range of aaRS?

To answer this question, we introduced additional information in the Results (specifically in the section on *in vitro* binding and bundling), as well as in the Discussion (first paragraph). We explain that actin bundlers can have K_d's that vary considerably, but many are in the micromolar range (~1-10 μM). As such, the individual molecules of bundlers can be coming on and off filament sides while the group of molecules collectively maintains a stable bundle. The K_d of YARS1^{WT} binding actin is 0.75 μM, the K_d of YARS1^{E196K} binding actin is 0.8 μM and are thus in the range of known established bundlers. The concentrations of F-actin and YARS1 used in our assays are fairly typical and were optimized to determine the K_d of YARS1 for F-actin. The concentration of F-actin we used is about two orders of magnitude below cellular concentrations of actin (~100 μM actin monomers, and >200 μM F-actin).

While we could not find data in the available literature reporting concentrations of YARS1 in eukaryotic cells, we found that an average copy number for the lysyl-tRNA synthetase (KARS1, also implicated in a subtype of CMT) is in the range of 10⁷ per cell (HeLa cell)¹⁶. The average copy number of actin in yeast cells is about an order of magnitude less (~1.5x10⁶)¹⁷, however the volume of a yeast cell is one order of magnitude smaller than the HeLa cell where KARS1 levels were determined. Thus, both actin and synthetases are a high copy number proteins and based on this estimate, synthetases (including YARS1), are probably in the range of tens of μM concentration in eukaryotic cells. Thus, the F-actin binding assays we conducted are likely in the range of physiologically relevant YARS1 concentrations.

Figure 2a and 2c should be separated and displayed next to their quantification – the current arrangement is rather confusing.

We have made the requested changes in Figure 2.

In light of the later findings, how does the identification of Coro, dPod, and IKKε as additional genetic modifiers fit? Are these expected to have overlapping functions with YARS1 or Fimbrin?

All three additional genetic modifiers we identified encode proteins with actin-binding and – bundling activities. Coro and its close homologue dPod1 (coronin family members) bind and bundle F-actin, and IKKε kinase is a regulator of F-actin bundling in flies. The dynamic nucleation, assembly, disassembly and bundling of F-actin is regulated by different actin-binding proteins (> 200 in humans). These actin-binding proteins collectively compete for overlapping binding sites on actin (in monomeric or polymeric forms) and influence specific steps in the formation, higher order organization, and turnover of actin networks. YARS1 with its actin-binding and bundling properties contributes to this cycle and likely competes with multiple established ABPs, some of which may be the genetic and physical interactors we identified. We have discussed this in the first paragraph of the Discussion.

S4: Was PLS3 the only identified Plastin due to its preferential expression by HEK293 cells? That should be easy to check using expression databases of HEK cells.

Indeed, PLS3 is about ten times more expressed than PLS1 in HEK293 cells according to The Human Protein Atlas (<https://www.proteinatlas.org/>) and this could be a reason we could detect only PLS3 as a binding partner of YARS1 in the IP of FLAG-YARS1^{WT} followed by mass

spectrometry. It is worth mentioning that the reciprocal experiment was also performed (BW, SB, personal communication), where PLS3 was immunoprecipitated followed by mass spectrometry and we retrieved YARS1 among the interactors, further supporting the interaction between the two proteins. As we demonstrate in the revised manuscript, the interaction between PLS3 and YARS1 is not direct, but it is likely via binding to F-actin. Furthermore, in the fly eye we identified interaction with both PLS3 and PLS2, albeit the ommatidial disorganization is milder in the background of PLS2 (see Figure 1 e, f).

Was actin itself found in the YARS1 interactome? The findings in Figure 2 suggest that it directly interacts with YARS1?

Yes, we did find actin in the YARS1 interactome in HEK293 cells. Please see the full list of YARS1 interactors in HEK293 cells as Supplemental Data 2. Please note that in addition to actin, we retrieved components and regulators of contractile actomyosin structures (non-muscle myosin II isoform A (NMMII), the actin crosslinker Filamin and the Rho-kinase), a variety of actin-binding proteins important for F-actin disassembly (Cofilin), control of filament growth (Capping protein). Translational proteins with established actin-binding properties and contributing to actin cytoskeletal organization were among the top YARS1 interactors too. These include the eukaryotic translation elongation factor 2 (eEF2) and the elongation factor eEF1A, listed as a YARS1 interactor in the BioGrid database¹⁸. eEF1A is known to bind and bundle F-actin¹⁹, and eEF2 has been identified in proteomic studies as a component of contractile actomyosin stress fibres²⁰. All of these proteins – including actin - were enriched in the YARS1-expressing HEK293 cells compared to the Flag-alone expressing control, so they are specific. Thus, the *in cellulo* interactomics places YARS1 in a network of proteins that are either *bona fide* components of the actin cytoskeleton or very strongly suggested to interact with actin. Of all these specific interactions we validated the direct binding between YARS1 and F-actin, while we could not establish physical interaction between YARS1 and PLS3 or alpha-actinin. We hypothesize that in the latter case the proteins are all bound to F-actin and are therefore retrieved together. While each of the remaining YARS1 interactors listed in Supplemental Data 2 needs to be validated in additional experiments, we believe that this summary is important to be published as it may be very useful to the field and to future studies in this area.

Figure 3: Figure 3 is clear, with informative schematics and intuitive data presentation. It would be helpful if this level of clarity could be achieved for Figure 1 and 2.

Based on Reviewer's suggestion, we added new schemes, or improved the existing schemes in Figures 1 and 2, and Supplementary Figure 5.

It is touched upon in the discussion but how do systemic changes in actin lead to the tissue-specific phenotype observed in CMT patients?

Increasing evidence associates actin-binding and regulatory proteins with degeneration of the neurons having the longest axons, e.g. the central and peripheral motor and sensory neurons. These are the most polarized cells in our body that are long-lived and particularly vulnerable to changes in their cytoskeleton. We gave specific examples of such associations in the new paragraph about actin cytoskeleton in the Introduction. While we cannot completely explain this specific

vulnerability, our findings support the idea that maintaining the dynamic equilibrium of actin cytoskeleton organization is crucial for the lifelong support of integrity and function of this neuronal population, as well as the nervous system as a whole.

Figure 4b: The figure legend should at least briefly describe the criteria used for quantification of Lifeact distribution or consider moving S6A to the main figure. In the provided images, the redistribution is more striking towards the inside of the bouton than close to the rim – how was the distribution at 700 nm chosen as a criterion?

We added more information to the legend of Figure 4 about the employed quantification procedure. Boutons at the larval NMJ are spherical to oval with a diameter between 2-5 μm and LifeAct intensity distributions vary from being restricted to the outermost region towards a broader diffuse rim or also covering the center of the bouton. Given the variation in bouton size and in intensity distributions between individual boutons and the light microscope resolution limit, we found that the most reliable and least ‘noisy’ parameter reflecting the redistribution from the outermost rim (200 nm) to a more diffuse distribution was obtained by taking the ratio with the intensity measured in close proximity, i.e. within a rim of (about) 700 nm. Parameters relying on intensity measurements inside the bouton were not able to capture the subtle changes in intensity distributions.

A question related to the actin cytoskeleton at the NMJ was also raised by Reviewer #3, To address this question, we have added new data showing the actin cytoskeleton (visualized alternatively with GFP-tagged Actin5C) at the NMJ in YARS1^{WT} and YARS1^{E196K} larvae. Please see Suppl. Fig 8 d, e. Furthermore, we used an unbiased approach to segment the actin assemblies and quantify their number in boutons. The results from this experiment support actin cytoskeleton rearrangements in the mutant YARS1 NMJs.

Figure 5: How can a similar rescue by Fimbrin in GARS mutants be explained if the suggested mode of action is through interaction between YARS1 and actin?

In the new panels of Suppl. Figure 5 i, j, we demonstrate direct interaction between GARS1 and actin. While the downstream effects from this binding remain to be delineated, the beneficial effect of manipulating the actin cytoskeleton on the neurotoxicity in the GARS1^{CMT} *Drosophila* model suggests that, in part, the observed global translational arrest and ISR response in the flies might be mediated by the actin cytoskeleton. Please see our discussion on that point in the 5th and 6th paragraphs of the Discussion, as well as our answer to question 2 of Reviewer 1.

YARS1 Mass spectrometry data should be deposited and made accessible after release (for example through PRIDE).

We have uploaded the raw mass spectrometry data to the PRIDE database under accession number PXD037630. In case the Reviewer would like to query the data, the account details are: **Username:** reviewer_pxd037630@ebi.ac.uk, **Password:** xYEU1C5M.

Page 6, missing word: We demonstrated that under stress conditions A fraction of ... (a missing).

This typo was adjusted in the revised version of the manuscript.

Reviewer #3 (Remarks to the Author):

In this paper, the authors identified *Drosophila* Fim (Fimbrin/CG8649) in an unbiased screen for enhancers of dominant mutations in the tyrosyl-tRNA synthetase (YARS1). This and other tRNA-ligases are mutated in Charcot-Marie-Tooth neuropathies (CMTs). Functionally, YARS1 exhibited an unexpected actin-binding and bundling activity, which suggests that misregulation of the actin cytoskeleton in mutant YARS1 proteins could be a driving force in neurodegeneration. While this hypothesis is compelling the data provided do not fully support this idea. I have therefore a number of questions and suggestions to improve the manuscript.

Major comments:

1) The authors screened 557 lines containing enhancer-promotor P-element insertions (EPs) and reference Rorth PNAS 1996 as a source (page 2). However, this paper mentions only 352 target lines (p. 12419). The M+M section reveals that the screened lines are derived from 4 different sources (page 14). This prevents precise tracking of the lines actually screened. The authors should therefore provide a complete list of the lines used for reproducibility purposes.

We appreciate the comment of the Reviewer that a complete list would be helpful for future readers, thus we are providing a Supplementary Data 1 file where we have listed all lines tested in this screen. The list contains the BDSC numbers, the full genotypes that sometimes correctly describe the gene they target. The putative genes targeted by the EPs were additionally manually curated. We refer to Rorth, PNAS, 1996²¹, as the first manuscript that describes the EP genomic elements and their implementation in genetic screens in general, and because we used the *EP* lines on the X chromosome in our genetic screen.

2) The authors state (p. 3) that they ultimately identified a single line BDSC_14274, which they call FimEP (p. 3) but sometimes also EPFim (Suppl. FigS1a). However, the Bloomington stock center lists CG5445 and not Fim as the parent gene under the accession number BDSC_14274. This should be clarified in the text, e.g. that the element is oriented to drive Fim but not CG5445. According to FlyBase, at the same location, there is also a second EP line, P(EP)FimG10929. Why was this line not detected in the screen? The authors should test if it behaves similar to BDSC_14274.

We are grateful to the Reviewer for noticing our inconsistent nomenclature used to refer to *BDSC_14274*. This line is now consistently referred to as *Fim^{EP}*. As suggested by the Reviewer, we clarified that *Fim^{EP}* is oriented to drive the expression of *Fim* and not *CG5445* in the main text where we first describe the interaction. In addition, once we validated the *BDSC_14274* as a YARS1^{E196K} modifier, we did test all publicly available EP lines including *P(EP)FimG10929* to confirm the interaction with an independent *EP* line. We did not detect ommatidial disorganization in the background of the *FimG10929*. Different EP lines are inserted at different locations in the *Fim* locus, and many of them are actually loss-of-function alleles, which might explain the lack of interaction with *FimG10929*. We detected only one more independent EP that induced mild ommatidial disorganization when co-expressed with YARS1^{E196K} – *Fim^{d02114}* or *Fim^{XP}* as referred

to in the revised manuscript. Scanning electron micrographs of eyes from flies expressing *Fim^{XP}* alone or together with YARS1 were added in Supplementary Figure 1d. Ultimately, we confirmed the interaction with transgenic UAS-Fimbrin flies.

3) Various effects of mutant YARS1 on actin organization is shown in different experiments. Fig. 2 shows that mutant YARS1 binds and bundles F-actin faster than wild-type YARS1 in biochemical assays. Fig. 3 shows perturbed F-actin cables and disrupted stress fibres in patient-derived fibroblasts in culture. Fig. 4 shows altered distribution of Lifeact::Ruby at *Drosophila* larval neuromuscular junctions (NMJs), and Fig. 5 shows YARS1-induced constrictions in the giant fiber system. All these findings are therefore individual observations in different systems, but it is unclear if they are causally connected.

Because of the dynamic nature of actin cytoskeleton, its complex regulation and its involvement in multitude of cellular processes, it is challenging to use only one experimental approach. The experimental paradigms we used were selected based on 1) their robustness, 2) relevance to disease, 3) our experience with these assays, and 4) the published examples from others about the application of similar assays in studying actin cytoskeletal defects. The genetic screen in the eye facilitated unbiased identification of a modifier, which in combination with further validation steps, pointed to a novel, unstudied function for YARS1. One of the prominent structures that determines ommatidial shape and function is the rhabdomere, formed by the photoreceptor neurons. Rhabdomeres are actually microvilli, filled with bundled actin, and thus the adult fly eye provided a high throughput screening platform for disease modifiers, and possibly facilitated the identification of interactors important in actin bundle formation/maintenance. The biochemical assays and the TIRF microscopy are specific *in vitro* approaches that demonstrated that YARS1 directly binds to F-actin and organizes actin filaments into higher order structures. In order to address synapse-specific defects that could be contributing to the neuropathy, we looked at established actin-dependent processes in the larval NMJ such as mobilization of synaptic vesicles. We feel that by providing independent evidence in different systems that point to the same conclusion about the novel property of YARS1 and the involvement of actin cytoskeleton in YARS1^{CMT} we made our statements stronger.

For example, are the constrictions in the giant fibre system caused by defects in the organization of F-actin, or are they caused by the observed defects in synaptic vesicle?

To address this important question, we looked at the terminals of *A307-Gal4>Fim^{EP}* flies, which have functional deficits, as depicted in Figure 6b. Neurobiotin dye-filling of the giant fibers in these flies showed abnormal terminals with thinning, constriction, and shortend teminals, like the defects observed in YARS1^{CMT} and GARS1^{CMT} mutants. The fact that an actin-binding protein causes similar defects to YARS1^{CMT} argues that they can arise directly from cytoskeletal disruption. We added this data to Supplementary Fig. 10c.

We also attempted visualizing the actin cytoskeleton in the Giant Fiber (GF) interneuron by expressing *UAS-Act5C::GFP* throughout the circuit (*A307-Gal4* driver). We isolated, fixed and stained the brain and the ventral nerve cord (VNC) from adult flies (GFP signal enhanced with anti-GFP nanobodies), and imaged with spinning disk confocal microscopy. We could only detect Act5C::GFP signal in GFs in the brain/VNC connection region (yellow rectangle and zoom-in),

and unsurprisingly, it looked homogeneously distributed with no specific structures to assess with the diffraction-limited microscopy we used. Furthermore, we could not detect Act5C::GFP in the GF terminals (expected in the region marked with dashed, red line). Thus, due to the limitations we encountered in this trial, we refrained from studying further actin cytoskeleton in this system, which will require improved actin markers, fixation procedures and microscopy with improved resolution.

Also, F-actin re-organization should lead to constrictions at NMJs, too, but synaptic boutons appear unchanged in Fig. 4a. In the end, it should be possible to generate CMT-like phenotypes, which are characterized by progressive distal muscle weakness.

We agree with the Reviewer that the representative synaptic boutons in Figure 4a (current Figure 5a) appear unchanged. In our previous study in Bervoets et al., Nat Comm 2019¹², where the same fly model was used, we demonstrated NMJ undergrowth defects at the examined muscles 6 and 7 in the YARS1^{CMT} expressing larvae. In addition, similar NMJ undergrowth and progressive denervation at a “distal” larval muscle (muscle 24), was described for GARS1^{CMT} expressing larvae in an independently generated *Drosophila* GARS1^{CMT} model in Niehues et al., Nat Comm 2015²². Thus, at least two CMT-causing synthetases affect the overall growth of larval NMJs in the fly models. We haven't noted any specific bouton defects, such as in size or shape. In general, manipulating actin binding proteins and F-actin reorganization could lead to a variety of morphological outcomes at the NMJ. As currently we cannot distinguish the specific contribution of F-actin reorganization to the NMJ growth defects in the YARS1^{CMT} NMJs, we assessed vesicle mobilization as a process well-known to depend on presynaptic F-actin changes (Figure 5).

4) For the interpretation of the detected genetic interaction between YARS1 and Fimbrin in the eye, it would be important to demonstrate that YARS1 and Fim also functionally co-operate in peripheral nerves and synergistically generate CMT-like phenotypes. While the authors show that

endogenous YARS1 localizes to synaptic boutons, the distribution of Fim is less clear, although several exon-trap lines seem to exist. Is endogenous Fim co-expressed with YARS1 in motoneurons or at NMJs? Would co-expression of Fim and YARS1 increase actin-bundling activity (biochemically, S2 cells), completely dissolve Lifeact::Ruby at NMJs or augment constrictions in the giant fibre system?

Endogenous Fim (Fim::GFP trap) and transgenic PLS3 (*nSyb-Gal4*-driven expression) are detected at the NMJ and we added additional panels with representative images in Supplemental Fig. 9e, f to demonstrate this localization.

We attempted to address the nature of the YARS1-PLS3 genetic interaction biochemically. We tested for direct interaction between the two proteins with two independent approaches (immunoprecipitation and a pull-down of recombinant proteins) and their possible competition on F-actin binding. As a result, we found no direct interaction between YARS1 and PLS3, and observed competitive rather than cooperative binding to F-actin at least at the three tested concentrations of YARS1. These data are added in Suppl. Figure 4c d, and Suppl. Figure 5d, e, respectively. They rather suggest a complex and possibly competitive regulation of actin cytoskeleton organization than a cooperative actin-bundling activity. A precise answer to this question will require additional, detailed analyses, as exemplified by the study of Audenhove et al., 2016²³, which used battery of assays to demonstrate that the two well established bundlers PLS2 and Fascin induce different type of bundles and their bundle characteristics are cooperatively employed to shape cancer-related structures like invadopodia and filopodia.

5) While the F-actin marker Lifeact::Ruby stains the "bouton border proximity" (Fig. 4), UAS-GFP-actin distributes in a defined punctate pattern at presynapses of larval NMJs (Pawson et al., JNS 28, 11111 (2008, there Fig. 3). Is there any explanation for this differential localization? Would YARS1 proteins also be able to re-localize GFP-actin and dissolve these puncta?

To address the Reviewer's comment, and in addition to the presented data with the UAS-Lifeact::Ruby actin marker, we co-expressed YARS1 (WT and mutant) and PLS3 with GFP-tagged actin5C (UAS-Act5C::GFP) to independently assess for actin rearrangements at the NMJ. Co-expression of GFP-tagged Actin (Act5C::GFP) with YARS1 and PLS3 demonstrated reduction of the number of presynaptic assemblies in boutons. Thus, independent use of actin and an actin marker demonstrated changes in the distribution and organization of presynaptic actin cytoskeleton upon YARS1^{CMT} expression, which aligns with the effects of the actin bundler PLS3. These new data are added in Suppl. Figure 8d, e in the revised manuscript, along with the experimental procedures in the Methods section.

6) The reason for seeding normal and patient-derived fibroblasts on Y-shaped microstructures rather than on normal tissue culture dishes is not explained (Fig. 3, p. 4, p.17).

Seeding the cells on micropatterns has the advantage of restricting the cell to adopt a predetermined shape by controlling adherence to the substrate while maintaining non-adhering compartments (the cell apices in the case of the Y-shaped micropatterns we have used). This adhesion pattern allows characteristic actin networks within the cell (prominent stress fibers at the cell periphery and branched actin network at the apices), which are more controlled

than when cells are seeded on regular glass or plastic culture dishes, and assume more diverse cell shapes and F-actin organization patterns. In this particular case, it is easier to detect deviations in the actin cytoskeleton organization induced by the mutant synthetase. We added the rationale for this approach in the Results. In addition, we included images of HeLa cells grown on standard substrate ('non-micropatterned') displaying the random and uncontrolled appearance of the actin cytoskeleton, which further support the need for spatial control of cell shape when comparing actin network morphology (Suppl. Figure 6).

Why should fibroblasts be informative if disease occurs in neurons and axons?

Even if patient-derived neurons are not available, it would be possible to transfect neurons with wild-type and mutant constructs and determine the consequences on F-actin organization.

To address the recommendation of the Reviewer, we studied human SH-SY5Y neuroblastoma cells stably expressing comparable levels of YARS1^{WT} or two CMT-causing mutations (YARS1^{E196K} or YARS1^{G41R}). Those cells are widely used for studying human neurodegenerative diseases. In their undifferentiated state, SH-SY5Y cells are rapidly dividing and have the characteristics of immature catecholaminergic neurons. They can be differentiated into a homogenous culture of stable neuron-like cells (SH-SY5Y-derived neurons). We employed a differentiation protocol known to induce SH-SY5Y-derived neurons showing characteristics typical for cholinergic neurons, like peripheral motoneurons²⁴. We first compared the global cell migration speed of undifferentiated cells, guided by the particular dependence on actin network remodeling necessary for protrusion dynamics, force generation and optimal cell motility²⁵, and by our observation of distinct YARS1 accumulations at the dynamic leading edges of HeLa cells (Suppl. Figure 6). In line with defects in actin cytoskeleton remodeling, we found significantly reduced global cell migration speeds in both YARS1^{G41R} and YARS1^{E196K} cells (Figure 4a, b). To confirm the involvement of protrusion dynamics in the slow migration of mutant cells, we performed faster phase-contrast microscopy at higher magnification and examined the formation and dynamics of individual lamellipodia, formed at the leading edge of migrating cells. Cells expressing YARS1^{G41R} and YARS1^{E196K} displayed smaller protrusions, which did not extend as far as in control cells (Figure 4c-e). Overall, these results show that YARS1^{CMT} prohibits the formation of fully persistent and extended lamellar protrusions and causes defects in cell migration. Next, we induced neuronal differentiation by treating the SH-SY5Y cells with retinoic acid, giving rise to the formation of axonal and dendritic structures (collectively referred to as neurites). Measurement of the overall number of neurites showed reduced neurite outgrowth in YARS1^{G41R} and YARS1^{E196K} compared to YARS1^{WT} cells (Figure 4f, g). Furthermore, specific assessment of individual neuronal cells revealed a significant decrease in the proportion of cells with secondary neurites and branching defects in YARS1^{CMT} neuron-like cells. Combined, these results show that YARS1^{CMT} causes delayed neurite outgrowth and neurite branching *in cellulo*. Ultimately, we studied the organization of the actin cytoskeleton *in vivo*, in the nervous system of *Drosophila*. We present new evidences that actin cytoskeleton is perturbed in the presynaptic actin assemblies in the boutons of the larval NMJ (Suppl. Figure 8d, e), as well as the functional consequences of actin cytoskeleton manipulations in the giant fiber circuit.

We studied the fibroblast cultures presented in Figure 3 because they represent an established platform for investigating actin cytoskeleton rearrangements and having the advantage of being isolated from one of the YARS1^{CMT} patients and expressing the wild type and mutant YARS1 at

endogenous levels. While the specific susceptibility of neurons to YARS1^{CMT} pathology might not be fully answered by using human fibroblasts, the obtained data provide guidance for the changes to look for in other (more relevant) systems, as we did in the SH-SY5Y cells. We believe the fibroblast cultures offer an opportunity to study the actin cytoskeletal perturbations in patient-derived iPSC-motoneurons in the future. However, this type of studies will require additional patient's samples, appropriate controls (e.g. isogenic controls) and improved cytological techniques which are not available now.

In addition, Fig. 3a and 3e show patient-derived fibroblasts but speckle-like accumulations, a characteristic phenotype of mutant YARS1, is visible only in Fig. 3a.

The images used to demonstrate the distribution of endogenous YARS1 and F-actin in former Figure 3e in the initial submission of the manuscript were single slices of the corresponding micrographs. To address the Reviewer's concern, we replaced the representative images with maximum intensity projections that facilitate visualizing all entities containing F-actin, including speckles in YARS1^{CMT} background. Please see new panel in Figure 3e.

Reduced co-localization with F-actin stress fibres in YARS1-mutant fibroblasts is not really visible in the images shown in Fig. 3e and might require higher magnifications (40x-100x objectives).

In the revised manuscript, in Figure 3e we introduce zoomed-in images of a stress fibre, an apex and speckles that enable better visualization of YARS1 and F-actin in the respective structures in the CMT-patient derived fibroblasts. In addition, we added new images showing the association of YARS1 and Lifeact in HeLa cells (Suppl. Figure 6 and Supplementary Movie 2).

Minor points:

a) Fig. 1: Insets are too small and should be enlarged, as it is not possible to judge bristle arrangement in printouts.

We addressed this point by changing the entire layout of Figure 1, where we improved the schemes, subdivided the eye micrographs into individual panels and enlarged the insets so that the arrangement of the ommatidia and bristles is better visualized.

In addition, Fig. 2f appears too dark.

We improved the contrast in Figure 2f, now Figure 2g.

b) Legend to Fig. 5 (page 14): panel "f" should be "e"

This typo was corrected in the revised manuscript in which this figure is numbered as Figure 6.

c) There are different classes of CMT. The authors should specify to which class they are referring in the paper (neuronal or glial forms).

In the revised Introduction we added that "the pathology is mostly restricted to the axons of the peripheral nerves" and thus we are dealing with CMT sub-forms that are predominantly axonal in nature.

d) Suppl. Fig. S2b: The meaning of the dashed line should be indicated in the legend.

In the legend, we now explain that the dashed line indicates the expression levels of the controls.

Suppl. Fig. S6: The control image in S6b is identical to the control image in Fig. 4a and should be replaced.

We stated in the legend of the initial Suppl. Figure 6b (Suppl. Figure S8b in the revised version) that we reused the control image from Figure 4a (currently Figure 5a), which facilitates the comparison of Lifeact distribution between different genotypes in the main and supplemental figures.

Suppl. Fig. S8 seems unnecessary.

In the revised version, we added another panel (panel c) that presents neurobiotin dye-filling of the GFs of adult flies expressing *Fim^{EP}* throughout the GF circuit (*A307-Gal4>Fim^{EP}*), important to support our claims that defects in actin organizing molecules, like actin bundlers, lead to neuronal defects.

Supp. Fig. S9 FasII-staining in S9c does not represent the average distribution of FasII at NMJs and should be replaced.

We replaced the image with a different representative one, which hopefully captures better the honeycomb-like distribution of FasII. Of note, the images in Suppl. Fig. S9 (current numbering Suppl. Fig S10) are single slices imaged with improved resolution (SIM microscopy). Thus, the discrete distribution of the assessed protein might be different from the more abundant diffraction-limited images found in previously published literature.

- 1 Wei, N. *et al.* Oxidative stress diverts tRNA synthetase to nucleus for protection against DNA damage. *Mol Cell* **56**, 323-332, doi:10.1016/j.molcel.2014.09.006 (2014).
- 2 Fu, G., Xu, T., Shi, Y., Wei, N. & Yang, X. L. tRNA-controlled nuclear import of a human tRNA synthetase. *J Biol Chem* **287**, 9330-9334, doi:10.1074/jbc.C111.325902 (2012).
- 3 Kyheröinen, S. & Vartiainen, M. K. Nuclear actin dynamics in gene expression and genome organization. *Semin Cell Dev Biol* **102**, 105-112, doi:10.1016/j.semcdb.2019.10.012 (2020).
- 4 Ermanoska, B. *et al.* CMT-associated mutations in glycyl- and tyrosyl-tRNA synthetases exhibit similar pattern of toxicity and share common genetic modifiers in *Drosophila*. *Neurobiol Dis* **68**, 180-189, doi:10.1016/j.nbd.2014.04.020 (2014).
- 5 Williams, T. D. & Rousseau, A. Actin dynamics in protein homeostasis. *Biosci Rep* **42**, doi:10.1042/bsr20210848 (2022).

- 6 Pakos-Zebrucka, K. *et al.* The integrated stress response. *EMBO Rep* **17**, 1374-1395, doi:10.15252/embr.201642195 (2016).
- 7 Silva, R. C., Sattlegger, E. & Castilho, B. A. Perturbations in actin dynamics reconfigure protein complexes that modulate GCN2 activity and promote an eIF2 response. *Journal of Cell Science* **129**, 4521-4533, doi:10.1242/jcs.194738 (2016).
- 8 Zuko, A. *et al.* tRNA overexpression rescues peripheral neuropathy caused by mutations in tRNA synthetase. *Science* **373**, 1161-1166, doi:10.1126/science.abb3356 (2021).
- 9 Sun, L. *et al.* CMT2N-causing aminoacylation domain mutants enable Nrp1 interaction with AlaRS. *Proc Natl Acad Sci U S A* **118**, doi:10.1073/pnas.2012898118 (2021).
- 10 He, W. *et al.* CMT2D neuropathy is linked to the neomorphic binding activity of glycyl-tRNA synthetase. *Nature* **526**, 710-714, doi:10.1038/nature15510 (2015).
- 11 Graziani, G. & Lacal, P. M. Neuropilin-1 as Therapeutic Target for Malignant Melanoma. *Front Oncol* **5**, 125, doi:10.3389/fonc.2015.00125 (2015).
- 12 Bervoets, S. *et al.* Transcriptional dysregulation by a nucleus-localized aminoacyl-tRNA synthetase associated with Charcot-Marie-Tooth neuropathy. *Nat Commun* **10**, 5045, doi:10.1038/s41467-019-12909-9 (2019).
- 13 Spaulding, E. L. *et al.* The integrated stress response contributes to tRNA synthetase-associated peripheral neuropathy. *Science* **373**, 1156-1161, doi:10.1126/science.abb3414 (2021).
- 14 Storkebaum, E. *et al.* Dominant mutations in the tyrosyl-tRNA synthetase gene recapitulate in *Drosophila* features of human Charcot-Marie-Tooth neuropathy. *Proc Natl Acad Sci U S A* **106**, 11782-11787, doi:10.1073/pnas.0905339106 (2009).
- 15 Ghosh, S. & Feany, M. B. Comparison of pathways controlling toxicity in the eye and brain in *Drosophila* models of human neurodegenerative diseases. *Hum Mol Genet* **13**, 2011-2018, doi:10.1093/hmg/ddh214 (2004).
- 16 David, A. *et al.* RNA binding targets aminoacyl-tRNA synthetases to translating ribosomes. *J Biol Chem* **286**, 20688-20700, doi:10.1074/jbc.M110.209452 (2011).
- 17 Wu, J. Q. & Pollard, T. D. Counting cytokinesis proteins globally and locally in fission yeast. *Science* **310**, 310-314, doi:10.1126/science.1113230 (2005).
- 18 Wan, C. *et al.* Panorama of ancient metazoan macromolecular complexes. *Nature* **525**, 339-344, doi:10.1038/nature14877 (2015).
- 19 Yang, F., Demma, M., Warren, V., Dharmawardhane, S. & Condeelis, J. Identification of an actin-binding protein from *Dictyostelium* as elongation factor 1a. *Nature* **347**, 494-496, doi:10.1038/347494a0 (1990).
- 20 Liu, S., Matsui, T. S., Kang, N. & Deguchi, S. Proteome of actin stress fibers. *bioRxiv*, 2021.2006.2001.446528, doi:10.1101/2021.06.01.446528 (2021).
- 21 Rørth, P. A modular misexpression screen in *Drosophila* detecting tissue-specific phenotypes. *Proc Natl Acad Sci U S A* **93**, 12418-12422, doi:10.1073/pnas.93.22.12418 (1996).
- 22 Niehues, S. *et al.* Impaired protein translation in *Drosophila* models for Charcot-Marie-Tooth neuropathy caused by mutant tRNA synthetases. *Nat Commun* **6**, 7520, doi:10.1038/ncomms8520 (2015).
- 23 Van Audenhove, I. *et al.* Fascin Rigidity and L-plastin Flexibility Cooperate in Cancer Cell Invadopodia and Filopodia. *J Biol Chem* **291**, 9148-9160, doi:10.1074/jbc.M115.706937 (2016).

- 24 Bell, M. & Zempel, H. SH-SY5Y-derived neurons: a human neuronal model system for investigating TAU sorting and neuronal subtype-specific TAU vulnerability. *Rev Neurosci* **33**, 1-15, doi:10.1515/revneuro-2020-0152 (2022).
- 25 Schaks, M., Giannone, G. & Rottner, K. Actin dynamics in cell migration. *Essays Biochem* **63**, 483-495, doi:10.1042/ebc20190015 (2019).

REVIEWERS' COMMENTS

Reviewer #1 (Remarks to the Author):

I appreciate the authors efforts in the revised version to explain the diverse molecular mechanisms of AARS. I felt that it would be useful to provide a simple summary cartoon on the different molecular functions of YARS1, different binding partners etc. which have been implicated in these defects and the suggestions how the variety of these different pathways interact with each other in shaping the disease.

The authors replied that a summary figure on the different molecular aspects of AARS CMT might be more suitable for a review article. However I think it would improve the understanding of the current manuscript to provide such a figure on YARS1.

Reviewer #2 (Remarks to the Author):

The authors have addressed all points that were raised and the manuscript is more comprehensive and accessible now. Great work!

Reviewer #3 (Remarks to the Author):

The authors have now substantially revised their manuscript and made great efforts to improve it. They now provide further evidence for a role of the Tyrosyl-tRNA Synthetase YARS in actin bundling and organization. In particular, they have now attached a complete list of lines used in their genetic modifier screen (Suppl. Data 1), which led to the identification of *Drosophila* Fimbrin (Fim), a known actin crosslinking protein, as a functional interaction partner of disease-related YARS proteins. In addition, they identified a second overexpression line in the *fim* locus that independently supports this synergism (Suppl. Fig. 1). Furthermore, thinning and gapping at synaptic endings of the giant fiber systems could be caused by rearrangements of the actin cytoskeleton (Suppl. Fig. 10).

The authors also extended their previous studies using fibroblasts to neuron-like cells (neuroblastoma cells), finding that mutant YARS proteins affect a variety of actin-controlled processes, such as neurite outgrowth, protrusion size or cellular migration speed (Figure 4). However, biochemical evidence for a direct interaction of recombinant Fim and YARS remains scarce, but immunoprecipitation experiments from human HEK293 cells expressing tagged wild-type YARS showed at least that Plastin 3, a homolog of Fim, could be detected in the precipitates using mass spectrometry (Suppl. Fig. 4, Suppl. Data 2). Similarly, spin-down experiments demonstrated that both proteins co-sedimented with actin filaments (Suppl. Fig. 5). These and other novel data have now been added to the manuscript and further support an unexpected role of YARS in bundling and organizing actin filaments.

Minor comments:

a) Suppl. Fig. 11: The reasoning for selecting the orientation of the plot line is not explained in the legend, i.e. it is not clear why the line in panel 11b projects along the lateral border of the bouton and not across the bouton like in all other examples in this figure. Please, change to a projection across the bouton, if possible.

RESPONSE TO REVIEWERS' COMMENTS

REVIEWERS' COMMENTS

Reviewer #1 (Remarks to the Author):

I appreciate the authors efforts in the revised version to explain the diverse molecular mechanisms of AARS. I felt that it would be useful to provide a simple summary cartoon on the different molecular functions of YARS1, different binding partners etc. which have been implicated in these defects and the suggestions how the variety of these different pathways interact with each other in shaping the disease.

The authors replied that a summary figure on the different molecular aspects of AARS CMT might be more suitable for a review article. However I think it would improve the understanding of the current manuscript to provide such a figure on YARS1.

Response: Following the Reviewer's suggestion, we provide a summary figure with the revised manuscript (please see the new Figure 7).

Reviewer #2 (Remarks to the Author):

The authors have addressed all points that were raised and the manuscript is more comprehensive and accessible now. Great work!

Response: We thank the Reviewer for his opinion. There were no further comments to address.

Reviewer #3 (Remarks to the Author):

The authors have now substantially revised their manuscript and made great efforts to improve it. They now provide further evidence for a role of the Tyrosyl-tRNA Synthetase YARS in actin bundling and organization. In particular, they have now attached a complete list of lines used in their genetic modifier screen (Suppl. Data 1), which led to the identification of *Drosophila* Fimbrin (Fim), a known actin crosslinking protein, as a functional interaction partner of disease-related YARS proteins. In addition, they identified a second overexpression line in the *fim* locus that independently supports this synergism (Suppl. Fig. 1). Furthermore, thinning and gapping at

synaptic endings of the giant fiber systems could be caused by rearrangements of the actin cytoskeleton (Suppl. Fig. 10).

The authors also extended their previous studies using fibroblasts to neuron-like cells (neuroblastoma cells), finding that mutant YARS proteins affect a variety of actin-controlled processes, such as neurite outgrowth, protrusion size or cellular migration speed (Figure 4). However, biochemical evidence for a direct interaction of recombinant Fim and YARS remains scarce, but immunoprecipitation experiments from human HEK293 cells expressing tagged wild-type YARS showed at least that Plastin 3, a homolog of Fim, could be detected in the precipitates using mass spectrometry (Suppl. Fig. 4, Suppl. Data 2). Similarly, spin-down experiments demonstrated that both proteins co-sedimented with actin filaments (Suppl. Fig. 5). These and other novel data have now been added to the manuscript and further support an unexpected role of YARS in bundling and organizing actin filaments.

Minor comments:

a) Suppl. Fig. 11: The reasoning for selecting the orientation of the plot line is not explained in the legend, i.e. it is not clear why the line in panel 11b projects along the lateral border of the bouton and not across the bouton like in all other examples in this figure. Please, change to a projection across the bouton, if possible.

Response: The reason the line in panel 11b projects along the lateral border of the bouton is to traverse along as many active zones as possible to assess their distribution relative to YARS1. We added a short description in the legend to navigate the readers better.